# Solving and Learning Partial Differential Equations with Variational Q-Exponential Processes

Guangting Yu    Shiwei Lan [*]

*School of Mathematical & Statistical Sciences*
*Arizona State University, Tempe, AZ 85287*

## Abstract

Solving and learning partial differential equations (PDEs) lies at the core of physics-informed machine learning. Traditional numerical methods, such as finite difference and finite element approaches, are rooted in domain-specific techniques and often lack scalability. Recent advances have introduced neural networks and Gaussian processes (GPs) as flexible tools for automating PDE solving and incorporating physical knowledge into learning frameworks. While GPs offer tractable predictive distributions and a principled probabilistic foundation, they may be suboptimal in capturing complex behaviors such as sharp transitions or non-smooth dynamics. To address this limitation, we propose the use of the $q$-exponential process (Q-EP), a recently developed generalization of GPs designed to better handle data with abrupt changes and to more accurately model derivative information. We advocate for Q-EP as a superior alternative to GPs in solving PDEs and associated inverse problems. Leveraging sparse variational inference, our method enables principled uncertainty quantification – a capability not naturally afforded by neural network-based approaches. Through a series of experiments, including the Eikonal equation, Burgers' equation, and an inverse Darcy flow problem, we demonstrate that the variational Q-EP method consistently yields more accurate solutions while providing meaningful uncertainty estimates.

**Keywords:** Probabilistic PDE Solvers, Bayesian Inverse Problems, Data Inhomogeneity, Modeling Derivatives, Uncertainty Quantification

## 1  Introduction

It is of fundamental importance in science and technology to solve mathematical models represented as a system of differential equations and to learn such a complex system by identifying crucial physical quantities with estimated uncertainty (a.k.a. inverse problems). Over centuries, theoretic foundations and computational methods have been developed for solving and learning partial differential equations (PDEs), which is facilitated by the development of modern computers. Traditional numerical algorithms such as the finite element method remain demanding for both domain knowledge and computing resources. Recently, there has been increasing interest and effort to efficiently automate this process using machine learning techniques.

The surge of physics-informed machine learning is driven by two main thrusts: neural network-based algorithms and Gaussian process (GP)-based probabilistic solvers. The former works are represented by physics-informed neural networks [PINN 37, 49], the deep Ritz method [9], the deep Galerkin method [44], and operator learning [20] methods including the Fourier neural operator [FNO 25], deep operator networks [DeepONet 26] and the neural inverse operator [NIO 28]. See [19] for a

---

[*]slan@asu.edu

review of recent advances. The basic idea is to parametrize the solution with a neural network and to minimize certain loss with respect to network parameters to obtain the solution. Despite empirical successes, these neural network-type approaches typically require large samples but lack convergence guarantee or uncertainty quantification (UQ). On the other hand, GP has been introduced to solve and learn ordinary differential equations (ODEs) [45, 39, 4, 43, 15] and PDEs [32, 33, 35, 38] with theoretic guarantee [34] and UQ [16]. More recent development on GP for solving and learning PDEs includes [6, 27, 14, 13, 29, 3]. Different from neural network approaches, these probabilistic solvers model the solution as a GP conditioned on PDE constraints and identify the solution as the maximum a posteriori (MAP).

Due to the tractability of conditional and predictive densities, GP has been widely adopted in machine learning and scientific computing [40]. However, as an $L_2$ regularization, GP tends to be over-smooth for modeling certain objects with abrupt changes or sharp contrast. For example, it is known in imaging analysis that GP may not detect or preserve edges very well in an image [23, 7]. On the other hand, researchers [42, 48] notice that total variation regularization penalizes the $L_1$ norm of derivatives and yields edge-preserving reconstructions. However, the total variation prior degenerates to GP prior with increasingly finer discretization mesh [23] and hence loses its edge-preserving feature. Therefore, [22] propose the Besov prior as an $L_q$ regularization and prove its discretization-invariant property. [24] further develop the $q$-exponential process (Q-EP) as a probabilistic definition of the Besov process with tractable posterior prediction and demonstrate it as a superior generalization of GP (with $q = 2$) in modeling inhomogeneous data with sharp transitions.

In this paper, we discover that Q-EP (with $q = 1$) is better in modeling derivative information than GP, and hence presents as a preferable candidate for solving PDEs. Heuristically, this is attributed to Q-EP's enhanced ability to model inhomogeneous objects with sharp variations, resulting in better regularization of large derivatives. Theoretically, this can be explained by a faster posterior convergence rate in Bayesian modeling with Q-EP priors. Unlike optimization-based approaches [6, 27], we adopt sparse variational inference [46, 47] for Q-EP [30, 5] to solve and learn PDEs, allowing natural UQ. An emerging challenge is that in addition to mapping the Q-EP mean function by the nonlinear PDE dynamics, one also needs to propagate the whole variational distribution through, which no longer renders a Q-EP. We solve this difficulty by linearizing the complicated PDE mapping. We also extend the resulting variational Q-EP solver for inverse problems.

**Connection to the literature**   Our work is motivated by [6] which optimizes the log-posterior for MAP as the PDE solution. Our proposed method replaces GP with a more general Q-EP and adopts variational Bayes for UQ. We investigate Q-EPs in solving various forward and inverse PDEs for a spectrum of $q$'s with $q = 2$ corresponding to GP. As a probabilistic solver, Q-EP may not be best compared with neural network-based approaches. However, we still include PINN [37] and Bayesian PINN [B-PINN 49] as baselines in our comparison. We emphasize that our algorithms rely only on limited data, e.g. boundary values or interior observations, while providing meaningful UQ. Our work is also related to the recently proposed physics-informed state-space GP [13], which however focuses on time-dependent PDEs. It adopts a variational spatiotemporal state-space GP and can be regarded as a related special case of ours for $q = 2$. Our work on solving and learning PDEs makes multiple contributions to the field of physics-informed machine learning:

1. It is a novel probabilistic PDE solver based on Q-EP with superior capability of modeling data inhomogeneity and derivative information.
2. It theoretically justifies the preference of Q-EP over GP in solving and learning PDEs.
3. It provides efficient UQ for solving forward and inverse PDE problems.

The remainder of the paper is organized as follows. Section 2 reviews Q-EP as a flexible prior in Bayesian models for inhomogeneous data and introduces an extension to incorporate derivative information. Section 3 explains the details of applying Q-EP to solve the forward and inverse problems of PDEs. We follow [6] to model the solution as MAP of Q-EP but highlight the challenges of variational inference including distribution propagation and variational lower bound. In Section 4, we justify the preference of Q-EP for $q = 1$ over GP ($q = 2$) in solving PDEs. In Section 5, we demonstrate the numerical advantages, particularly faster convergence, of Q-EP compared with alternatives using forward problems involving Eikonal equation and Burgers' equation and inverse problems of identifying permeability in the Darcy flow. Section 6 concludes with a discussion of limitations and future improvements.

## 2 Bayesian Modeling with Q-Exponential Process

### 2.1 $Q$-Exponential Process

The univariate *q-exponential distribution* [7] has density $\pi_q(u) \propto \exp\left(-\frac{1}{2}|u|^q\right)$, whose logarithm yields an $L_q$ regularization term. [24] generalize the univariate $q$-exponential random variable to a multivariate random vector, based on which a stochastic process can be defined. Suppose a function $u(\mathbf{x})$ is observed at $N$ locations, $\mathbf{x}_1, \cdots, \mathbf{x}_N \in \Omega \subset \mathbb{R}^d$. [24] define the *multivariate q-exponential distribution* for $\mathbf{u} := (u(\mathbf{x}_1), \cdots, u(\mathbf{x}_N))$, as a member of the family of elliptic distributions [18].

**Definition 1.** *A multivariate random vector $\mathbf{u} \in \mathbb{R}^N$ follows the q-exponential distribution, denoted as $\mathbf{u} \sim$ q-ED$_N(\boldsymbol{\mu}, \mathbf{C})$, if it has the following density:*

$$p(\mathbf{u}|\boldsymbol{\mu}, \mathbf{C}, q) = \frac{q}{2}(2\pi)^{-\frac{N}{2}}|\mathbf{C}|^{-\frac{1}{2}}r(\mathbf{u})^{(\frac{q}{2}-1)\frac{N}{2}}\exp\left\{-\frac{r^{\frac{q}{2}}}{2}\right\}, \quad r = (\mathbf{u}-\boldsymbol{\mu})^\mathsf{T}\mathbf{C}^{-1}(\mathbf{u}-\boldsymbol{\mu}). \quad (1)$$

**Remark 1.** *The negative log density of* q-ED *in* (1) *yields a quantity dominated by some weighted $L_q$ norm of $\mathbf{u} - \boldsymbol{\mu}$, i.e. $\frac{1}{2}r^{\frac{q}{2}} = \frac{1}{2}\|\mathbf{u}-\boldsymbol{\mu}\|_\mathbf{C}^q$. From the optimization perspective,* q-ED*, when used as a prior, imposes $L_q$ regularization in obtaining the maximum a posteriori (MAP).*

Li et al. [24] prove that the above multivariate $q$-exponential distribution satisfies the conditions of Kolmogorov's extension theorem [31] and thus can be generalized to a stochastic process. For this purpose, we scale $\mathbf{u} \sim$ q-ED$_N(\mathbf{0}, \mathbf{C})$ by a factor $N^{\frac{1}{2}-\frac{1}{q}}$ so that the *scaled q-exponential random variable* $\mathbf{u}^* := N^{\frac{1}{2}-\frac{1}{q}}\mathbf{u} \sim$ q-ED$_N^*(\mathbf{0}, \mathbf{C})$ has covariance asymptotically finite [Proposition 3.1 of 24]. With a covariance (symmetric and positive-definite) kernel $\mathcal{C} : \Omega \times \Omega \rightarrow \mathbb{R}$, we define the following *q-exponential process (Q-EP)* based on the scaled $q$-exponential distribution.

**Definition 2.** *A (centered) q-exponential process $u(\mathbf{x})$ with a kernel $\mathcal{C}$,* q-$\mathcal{EP}(0, \mathcal{C})$*, is a collection of random variables such that any finite set, $\mathbf{u} = (u(\mathbf{x}_1), \cdots u(\mathbf{x}_N))$, follows a scaled multivariate q-exponential distribution* q-ED$^*(\mathbf{0}, \mathbf{C})$*, where $\mathbf{C} = [\mathcal{C}(\mathbf{x}_i, \mathbf{x}_j)]_{N \times N}$.*

**Remark 2.** *When $q = 2$,* q-ED$_N(\boldsymbol{\mu}, \mathbf{C})$ *reduces to $\mathcal{N}_N(\boldsymbol{\mu}, \mathbf{C})$ and* q-$\mathcal{EP}(0, \mathcal{C})$ *becomes $\mathcal{GP}(0, \mathcal{C})$. When $q \in (0, 2)$,* q-$\mathcal{EP}(0, \mathcal{C})$ *lends flexibility to modeling functional data with more regularization than GP. In practice, $q = 1$ is often adopted for faster posterior convergence [1, 21] and the capability of preserving inhomogeneous features (rough functional data, edges in image, etc).*

The covariance kernel $\mathcal{C}$ is associated with a Hilbert-Schmidt (HS) integral operator $T_\mathcal{C} : L^2(\Omega) \rightarrow L^2(\Omega), u(\cdot) \mapsto \int_\Omega \mathcal{C}(\cdot, \mathbf{x}')u(\mathbf{x}')\mu(d\mathbf{x}')$ which has eigen-pairs $\{\lambda_\ell, \phi_\ell(\cdot)\}_{\ell=1}^\infty$ such that for $\forall \ell \in \mathbb{N}$, $T_\mathcal{C}\phi_\ell(\mathbf{x}) = \phi_\ell(\mathbf{x})\lambda_\ell$ and $\|\phi_\ell\|_2 = 1$. Assume $T_\mathcal{C}$ is trace-class, i.e. $\text{tr}(T_\mathcal{C}) := \sum_{\ell=1}^\infty \lambda_\ell < \infty$. Theorem 3.4 of [24] presents a series representation of Q-EP similar to GP and the Besov process [8].

**Theorem 2.1** (Karhunen-Loéve). *If $u(\cdot) \sim$ q-$\mathcal{EP}(0, \mathcal{C})$ with a trace-class HS operator $T_\mathcal{C}$ having eigen-pairs $\{\lambda_\ell, \phi_\ell(\cdot)\}_{\ell=1}^\infty$, then we have the following series representation for $u(\mathbf{x})$:*

$$u(\mathbf{x}) = \sum_{\ell=1}^\infty u_\ell \phi_\ell(\mathbf{x}), \quad u_\ell := \int_\Omega u(\mathbf{x})\phi_\ell(\mathbf{x}) \overset{ind}{\sim} \text{q-ED}^*(0, \lambda_\ell), \quad (2)$$

*where $\text{E}[u_\ell] = 0$ and $\text{Cov}(u_\ell, u_{\ell'}) = \lambda_\ell \delta_{\ell\ell'}$ with Dirac function $\delta_{\ell\ell'} = 1$ if $\ell = \ell'$ and $0$ otherwise. Moreover, we have $\text{E}[\|u(\cdot)\|_2^2] = \sum_{\ell=1}^\infty \text{E}[u_\ell^2] = \text{tr}(T_\mathcal{C}) < \infty$.*

### 2.2 Bayesian Regression with Q-EP Priors

Given $\mathbf{X}_{N \times d} = \{\mathbf{x}_n\}_{n=1}^N$ and $\mathbf{y}_{N \times 1} = \{y_n\}_{n=1}^N$, we consider the Bayesian regression model:
$$\begin{aligned}\mathbf{y} &= u(\mathbf{X}) + \boldsymbol{\varepsilon}, \quad \boldsymbol{\varepsilon} \sim \text{q-ED}_N(0, \Gamma), \\ u &\sim \text{q-}\mathcal{EP}(0, \mathcal{C}).\end{aligned} \quad (3)$$

Li et al. [24, Theorem 3.5] show that the posterior (predictive) distribution is analytically tractable when both the prior and the likelihood are $q$-exponential.

**Theorem 2.2.** *For the regression model* (3)*, the posterior distribution of $u(\mathbf{x}_*)$ at $\mathbf{x}_*$ is*
$$u(\mathbf{x}_*)|\mathbf{y}, \mathbf{X}, \mathbf{x}_* \sim \text{q-ED}(\boldsymbol{\mu}^*, \mathbf{C}^*),$$
$$\boldsymbol{\mu}^* = \mathbf{C}_*^\mathsf{T}(\mathbf{C}+\Gamma)^{-1}\mathbf{y}, \ \mathbf{C}^* = \mathbf{C}_{**} - \mathbf{C}_*^\mathsf{T}(\mathbf{C}+\Gamma)^{-1}\mathbf{C}_*,$$
*where $\mathbf{C} = \mathcal{C}(\mathbf{X}, \mathbf{X})$, $\mathbf{C}_* = \mathcal{C}(\mathbf{X}, \mathbf{x}_*)$, and $\mathbf{C}_{**} = \mathcal{C}(\mathbf{x}_*, \mathbf{x}_*)$.*

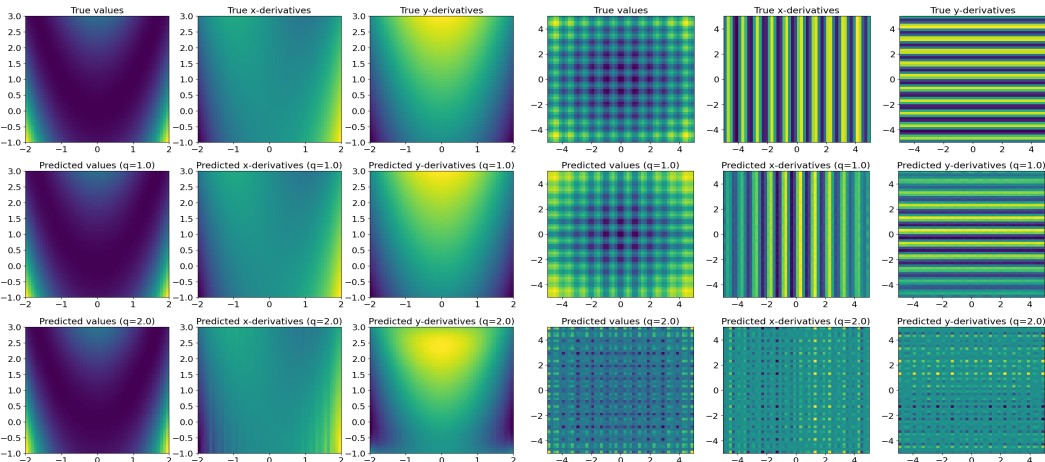

Figure 1: Contrasting Q-EP ($q = 1.0$, middle row) with GP ($q = 2.0$, bottom row) against the truth (top row) for modeling function values and derivatives of Rosenbrock (left) and Rastrigin (right).

## 2.3 Modeling with Derivative Information

Let $u \sim$ q-$\mathcal{EP}(0, \mathcal{C})$. Denote the function and its derivatives by $\tilde{u} = (u, \frac{\partial}{\partial \mathbf{x}} u, \cdots, \frac{\partial^k}{\partial \mathbf{x}^k} u)$ up to order $k$. Because linear operation preserves elliptic distributions [18, 10], $\tilde{u} \sim$ q-$\mathcal{EP}(0, \tilde{\mathcal{C}})$ is also a Q-EP if $\mathcal{C}$ in Definition 2 is differentiable up to order $k$, where the augmented kernel, $\tilde{\mathcal{C}}$, has a structure illustrated in Table A.1. For example, the $(1, 2)$ block of $\tilde{\mathcal{C}}$ is interpreted as $\mathrm{Cov}(u(\mathbf{x}), \frac{\partial}{\partial \mathbf{x}'} u(\mathbf{x}')) = \frac{\partial}{\partial \mathbf{x}'} \mathcal{C}(\mathbf{x}, \mathbf{x}')$. For solving the second-order PDEs in Section 5, we could adopt Matérn kernel (`matern52` for $\nu = 5/2$), $\mathcal{C}(\mathbf{x}, \mathbf{x}') = \sigma^2(1 + \sqrt{5}r + \frac{5}{3}r^2) \exp(-\sqrt{5}r)$, $r = \sqrt{\sum_{i=1}^d (x_i - x_i')^2 / \rho_i^2}$, for which the process is twice differentiable in the mean-square sense.

To prepare for solving PDEs, we define $\| \cdot \|_{s,q}$ for $u$ based on (2) with a smoothness parameter $s > 0$ and an integrability parameter $q \geq 1$ [22, 7]: $\|u(\cdot)\|_{s,q} = \left( \sum_{\ell=1}^\infty \ell^{\tau_q(s)q} |u_\ell|^q \right)^{\frac{1}{q}}$, $\tau_q(s) = \frac{s}{d} + \frac{1}{2} - \frac{1}{q}$. Consider the Banach space $B^{s,q}(\Omega) := \{u : \Omega \to \mathbb{R} \,|\, \|u(\cdot)\|_{s,q} < \infty\}$. If $q = 2$ and $\{\phi_\ell\}_{\ell=1}^\infty$ form the Fourier basis, then $B^{s,2}(\Omega)$ reduces to the Sobolev space $H^s(\Omega)$. For $u$ to be regular enough, we make the assumption on $\tilde{\mathcal{C}}$ so that $\tilde{u} \in L^q(\Omega)$ almost surely by the following proposition [21].

**Assumption 1.** *Suppose $\lambda = \{\lambda_\ell\}_{\ell=1}^\infty$ are eigenvalues of HS operator $T_{\tilde{\mathcal{C}}}$ for the kernel $\tilde{\mathcal{C}}$. We assume $\lambda \in \ell^{\frac{q}{2}}$, i.e. $\|\lambda\|_{\frac{q}{2}}^{\frac{q}{2}} = \sum_{\ell=1}^\infty \lambda_\ell^{\frac{q}{2}} < \infty$.*

**Proposition 2.1.** *If $\tilde{u}(\cdot) \sim$ q-$\mathcal{EP}(0, \tilde{\mathcal{C}})$ with a trace-class HS operator $T_{\tilde{\mathcal{C}}}$ satisfying Assumption 1, then $\tilde{u}(\cdot) \in L^q_{\mathbb{P}}(\mathbb{R}^\infty, L^q(\Omega)) := \{\tilde{u} : \Omega \times \mathbb{R}^\infty \to \mathbb{R} | \mathbb{E}(\|\tilde{u}\|_q^q) < \infty\}$ and $\mathbb{E}[\|\tilde{u}(\cdot)\|_q^q] = \|\lambda\|_{\frac{q}{2}}^{\frac{q}{2}} < \infty$.*

*Proof.* See Appendix B. $\qquad\square$

Let $\tilde{\mathbf{U}} := \tilde{u}(\mathbf{X})_{N \times (1+kd)} = [u(\mathbf{X}), \frac{\partial}{\partial \mathbf{x}} u(\mathbf{X}), \cdots, \frac{\partial^k}{\partial \mathbf{x}^k} u(\mathbf{X})]$. For brevity, we denote $D = 1 + kd$. Then $\mathbf{Y}_{N \times D}$ and $\mathbf{E}_{N \times D}$ are the corresponding observations and errors respectively in the model (3). We apply this model to Rosenbrock ($f(\mathbf{x}) = \sum_{i=1}^d [100(x_{i+1} - x_i^2)^2 + (1 - x_i)^2]$) and Rastrigin ($f(\mathbf{x}) = 10d + \sum_{i=1}^d [x_i^2 - 10 \cos(2\pi x_i)]$) test functions with function values and their derivatives observed on a $20 \times 20$ grid, i.e. $N = 400, d = 2, k = 1$. Figure 1 contrasts Q-EP ($q = 1.0$) and GP ($q = 2.0$) in predicting function and derivative values on the $50 \times 50$ grid. Q-EP outperforms GP in yielding a more accurate recovery, especially in the more challenging example of Rastrigin function.

Heuristically, the superiority of Q-EP in modeling derivatives over GP comes from its improved ability to handle inhomogeneous data with sharp variation. From the perspective of the $L_q$ norm of the gradient, the $L_1$ norm imposes stronger regularization than the $L_2$ norm on large values in $\nabla_{\mathbf{x}} f$,

resulting in a better model for prediction. In Section 4 we will give a more rigorous justification. In the following, we take advantage of Q-EP's capability of modeling derivative information and apply it to solve PDEs.

# 3 Solving Partial Differential Equations with Q-EP

## 3.1 Bayesian Solver

Consider the following general PDE defined on a bounded domain $\Omega \subset \mathbb{R}^d$:

$$
\begin{aligned}
\mathcal{D}(u)(\mathbf{x}) &= f(\mathbf{x}), \quad \mathbf{x} \in \Omega, \\
\mathcal{B}(u)(\mathbf{x}) &= g(\mathbf{x}), \quad \mathbf{x} \in \partial\Omega.
\end{aligned}
\tag{4}
$$

where $\mathcal{D} : B^{s,q}(\Omega) \to L^q(\Omega)$ is a differential operator and $\mathcal{B} : B^{s,q}(\partial\Omega) \to L^q(\partial\Omega)$ is a boundary operator with data $f \in L^q(\Omega)$ and $g \in L^q(\partial\Omega)$. Here we assume a sufficiently large smoothness index $s > 0$ such that the PDE (4) is well-defined pointwise and has a unique strong solution [6]. Let $\overline{\Omega} = \Omega \cup \partial\Omega$. For convenience of exposition, we denote the joint operator as $\mathcal{P} = (\mathcal{D}, \mathcal{B}) : B^{s,q}(\overline{\Omega}) \to L^q(\overline{\Omega})$, and the right-hand side function as $h = (f, g) \in L^q(\overline{\Omega})$.

A set of collocation points $\mathbf{X} = \{\mathbf{x}_n\}_{n=1}^N$ consists of $N_d$ interior points $\mathbf{X}_d = \{\mathbf{x}_1, \cdots, \mathbf{x}_{N_d} \in \Omega\}$ and $N_b$ boundary points $\mathbf{X}_b = \{\mathbf{x}_{N_d+1}, \cdots, \mathbf{x}_N \in \partial\Omega\}$, i.e. $\mathbf{X} = \mathbf{X}_d \cup \mathbf{X}_b$, and $N = N_d + N_b$. Regarding the evaluation of $\mathcal{P}(u)$ on $\mathbf{X}$, we make the following assumption so that we can properly define the likelihood model.

**Assumption 2.** *There exists a differentiable function $P : \mathbb{R}^D \to \mathbb{R}$ such that $\mathcal{P}(u)(\mathbf{x}) = P(\tilde{u}(\mathbf{x}))$. And further there is a constant $C > 0$ such that $\|\nabla P\| \leq C$.*

Then $\mathcal{P}(u)(\mathbf{X})$ becomes a nonlinear function of $\tilde{u}(\mathbf{X})$, denoted as $P(\tilde{u}(\mathbf{X})) = \mathcal{P}(u)(\mathbf{X})$. Let $\mathbf{h} = h(\mathbf{X})$. The probabilistic solver seeks to obtain $\tilde{\mathbf{U}} = \tilde{u}(\mathbf{X})$ based on observations $(P(\tilde{u}(\mathbf{X})), \mathbf{h})$.

Even if we model $\tilde{u}(\mathbf{X}) \sim \text{q-ED}(\tilde{\mathbf{u}}, \mathbf{S})$ with $\tilde{\mathbf{u}}, \mathbf{S}$ to be specified in (8) in Section 3.2, the nonlinear mapping $P$ would not render $P(\tilde{u}(\mathbf{X}))$ another q-ED random variable. Therefore, to properly define the likelihood model, we propose the following *distribution propagation* by linearizing $P$:

$$
\begin{aligned}
P(\tilde{u}(\mathbf{X})) &\approx P(\tilde{\mathbf{u}}_0) + \nabla P(\tilde{\mathbf{u}}_0)(\tilde{u}(\mathbf{X}) - \tilde{\mathbf{u}}_0) \sim \text{q-ED}(\mathbf{m}, \boldsymbol{\Gamma}), \\
\mathbf{m} &= P(\tilde{\mathbf{u}}_0) + \nabla P(\tilde{\mathbf{u}}_0)(\tilde{\mathbf{u}} - \tilde{\mathbf{u}}_0), \quad \boldsymbol{\Gamma} = \nabla P(\tilde{\mathbf{u}}_0)\mathbf{S}\nabla P(\tilde{\mathbf{u}}_0)^\mathsf{T} + \delta \mathbf{I}_N.
\end{aligned}
\tag{5}
$$

where the Taylor expansion of $P$ is about $\tilde{\mathbf{u}}_0$, which can be chosen as $\tilde{\mathbf{u}}_{n-1}$ from the previous training epoch or simply $\tilde{\mathbf{u}}$, and $\delta > 0$ is a small nugget to ensure positive definiteness of $\boldsymbol{\Gamma}$.

Let $\mathbf{Y}_{N \times 1} := P(\tilde{u}(\mathbf{X}))$. The Q-EP solver aims to minimize the discrepancy $\mathbf{E} = \mathbf{Y} - \mathbf{h}$. The potential (negative log-likelihood) function, $\Phi : L^q(\Omega) \times \mathbb{R}^N \to \mathbb{R}$, can then be defined:

$$
\Phi(\tilde{u}; \mathbf{Y}) = -\varphi(r; \boldsymbol{\Gamma}, N), \quad r = (\mathbf{m} - \mathbf{h})^\mathsf{T} \boldsymbol{\Gamma}^{-1}(\mathbf{m} - \mathbf{h}).
\tag{6}
$$

where $\varphi(r; \boldsymbol{\Gamma}, N) := -\frac{1}{2}\log|\boldsymbol{\Gamma}| + \frac{N}{2}\left(\frac{q}{2} - 1\right)\log r - \frac{1}{2}r^{\frac{q}{2}}$. Under Assumption 2, $\Phi$ is Lipschitz continuous in $\tilde{u}$, which is used in the convergence theorem in Section 4.

**Proposition 3.1.** *Suppose that the PDE mapping $P$ satisfies Assumption 2. Let $q \in (0, 2]$. Then for every $r > 0$, there exists $L = L(r) > 0$ such that for every $\mathbf{Y} \in \mathbb{R}^N$ and for all $\tilde{u}_1, \tilde{u}_2 \in L^q(\Omega)$ with $\max\{\|\tilde{u}_1\|_q, \|\tilde{u}_2\|_q\} < r$, $|\Phi(\tilde{u}_1; \mathbf{Y}) - \Phi(\tilde{u}_2; \mathbf{Y})| \leq L\|\tilde{u}_1 - \tilde{u}_2\|_q$.*

*Proof.* See Appendix B. □

Therefore, the Bayesian model for the solution $u$ to (4) can be summarized as

$$
\begin{aligned}
\mathbf{Y}|\tilde{u}(\mathbf{X}), \mathbf{h} &\sim \text{q-ED}_N(\mathbf{h}, \boldsymbol{\Gamma}), \\
\tilde{u} &\sim \text{q-}\mathcal{EP}(0, \tilde{\mathcal{C}}).
\end{aligned}
\tag{7}
$$

Our goal is to infer the posterior $p(\tilde{\mathbf{U}}|\mathbf{Y}) \propto p(\mathbf{Y}|\tilde{\mathbf{U}})p(\tilde{\mathbf{U}})$. Note that because the extended function $\tilde{u}(\mathbf{X})$ enters the likelihood model in a nonlinear way, Theorem 2.2 does not apply. In the following, we solve the inference problem using variational Bayes.

## 3.2 Variational Inference

We approximate the posterior $p(\tilde{\mathbf{U}}|\mathbf{Y})$ with some variational distribution $q(\tilde{\mathbf{U}})$ using the variational Bayes method, which aims to minimize the Kullback–Leibler divergence $\mathrm{KL}(q(\tilde{\mathbf{U}})\|p(\tilde{\mathbf{U}}|\mathbf{Y}))$. Because $\log p(\mathbf{Y}) = \mathrm{KL}(q(\tilde{\mathbf{U}})\|p(\tilde{\mathbf{U}}|\mathbf{Y})) + \mathcal{L}(q(\tilde{\mathbf{U}}))$, it reduces to maximizing the lower bound $\mathcal{L}(q(\tilde{\mathbf{U}}))$. The sparse variational approximation [46, 47] is adopted by introducing inducing points $\tilde{\mathbf{X}} \in \mathbb{R}^{M \times d}$ with their function values $\tilde{\mathbf{V}} = \tilde{u}(\tilde{\mathbf{X}}) \in \mathbb{R}^{M \times D}$.

With the variational distribution for inducing values $q(\tilde{\mathbf{V}}) \sim \text{q-ED}_{MD}(\boldsymbol{\mu}, \boldsymbol{\Sigma})$, the marginal variational distribution $q(\tilde{\mathbf{U}})$ can be obtained as [17, 30]

$$
\begin{aligned}
q(\tilde{\mathbf{U}}) = \int q(\tilde{\mathbf{U}}, \tilde{\mathbf{V}})d\tilde{\mathbf{V}} = \int p(\tilde{\mathbf{U}}|\tilde{\mathbf{V}})q(\tilde{\mathbf{V}})d\tilde{\mathbf{V}} \sim \text{q-ED}(\tilde{\mathbf{u}}, \mathbf{S}), \\
\tilde{\mathbf{u}} = \tilde{\mathbf{C}}_{NM}\tilde{\mathbf{C}}_{MM}^{-1}\mathrm{vec}(\boldsymbol{\mu}), \quad \mathbf{S} = \tilde{\mathbf{C}}_{NN} + \tilde{\mathbf{C}}_{NM}\tilde{\mathbf{C}}_{MM}^{-1}(\boldsymbol{\Sigma} - \tilde{\mathbf{C}}_{MM})\tilde{\mathbf{C}}_{MM}^{-1}\tilde{\mathbf{C}}_{MN}.
\end{aligned}
\tag{8}
$$

The final evidence lower bound (ELBO) $\mathcal{L}^*(q(\tilde{\mathbf{U}}))$ is (Refer to Section A.1 for more details.)

$$
\begin{aligned}
\mathcal{L}^*(q(\tilde{\mathbf{U}})) =& \varphi(\tilde{\mathbf{u}}^{\mathsf{T}}\boldsymbol{\Gamma}^{-1}\tilde{\mathbf{u}} + \mathrm{tr}(\boldsymbol{\Gamma}^{-1}\mathbf{S}); \boldsymbol{\Gamma}, N) \\
&+ \frac{1}{2}\log|\boldsymbol{\Sigma}| + \varphi(\langle\mathrm{vec}(\boldsymbol{\mu})^{\mathsf{T}}\tilde{\mathbf{C}}_{MM}^{-1}\mathrm{vec}(\boldsymbol{\mu})\rangle + \mathrm{tr}(\boldsymbol{\Sigma}\tilde{\mathbf{C}}_{MM}^{-1}); \tilde{\mathbf{C}}_{MM}, MD).
\end{aligned}
\tag{9}
$$

The variational solution $q(\tilde{\mathbf{U}})$ can be obtained by maximizing the ELBO (9) with respect to the variational parameters $(\boldsymbol{\mu}, \boldsymbol{\Sigma}, \tilde{\mathbf{X}})$ and hyper-parameters in the kernel $\mathcal{C}$. By introducing the $M$ inducing points, the overall computational complexity is reduced from $\mathcal{O}(N^3)$ to $\mathcal{O}(NM^2)$ [47].

## 3.3 Bayesian Inverse Problems

The above Bayesian framework can be readily extended to solve inverse problems. The following adaptation enables us to obtain both forward and inverse PDE solutions simultaneously.

Suppose that the PDE (4) contains a quantity of interest, $a(\mathbf{x})$, which could appear in the differential equation $\mathcal{D}$ or as part of the boundary condition $\mathcal{B}$. The task of Bayesian inverse problems is to find a true solution, $a^\dagger$, with proper UQ based on observations. Suppose $a$ is differentiable enough and we denote $\tilde{a} = (a, \frac{\partial}{\partial\mathbf{x}}a, \cdots, \frac{\partial^{k'}}{\partial\mathbf{x}^{k'}}a)$ to the order $k' \leq k$. Now the joint operator $\mathcal{P}$ applies to both $u$ and $a$, which produces a nonlinear function of $\tilde{u}(\mathbf{X}), \tilde{a}(\mathbf{X})$ when evaluated on $\mathbf{X}$, denoted as $P(\tilde{u}(\mathbf{X}), \tilde{a}(\mathbf{X})) = \mathcal{P}(u, a)(\mathbf{X})$.

In addition, there is an observation operator $\mathcal{O}$ such that observations, $\mathcal{O}(u)(\mathbf{X}) = u(\mathbf{X}_o)$, are obtained on some set of $N_o$ observation points, $\mathbf{X}_o \subset \overline{\Omega}$ with $|\mathbf{X}_o| = N_o$. This can be achieved by solving (4) with true $a^\dagger$ in simulations or simply modeling measurement data as noisy realization of (4) in real-world applications. Let $\tilde{N} = N + N_o$. We supplement the joint equation operator $\mathcal{P}$ with the observation operator $\mathcal{O}$ to form an augmented operator $\tilde{\mathcal{P}} = (\mathcal{P}, \mathcal{O})$. Therefore, we have $\tilde{\mathbf{Y}}_{\tilde{N}\times 1} = \tilde{\mathcal{P}}(u, a)(\mathbf{X}) = \tilde{P}(\tilde{u}(\mathbf{X}), \tilde{a}(\mathbf{X})) = [P(\tilde{u}(\mathbf{X}), \tilde{a}(\mathbf{X}))^{\mathsf{T}}, O(\tilde{u}(\mathbf{X}))^{\mathsf{T}}]^{\mathsf{T}}$. Similarly, we augment the right-hand side data $\mathbf{h}$ with observed $u(\mathbf{X}_o)$ to make $\tilde{\mathbf{h}}_{\tilde{N}\times 1} = [\mathbf{h}^{\mathsf{T}}, u(\mathbf{X}_o)^{\mathsf{T}}]^{\mathsf{T}}$.

If we model $\tilde{a}(\mathbf{X})$ using $q(\tilde{a}(\mathbf{X})) \sim \text{q-ED}(\tilde{\mathbf{a}}, \mathbf{S}_a)$ with variational mean and covariance $\tilde{\mathbf{a}}, \mathbf{S}_a$ respectively, then $q(\tilde{u}(\mathbf{X}))q(\tilde{a}(\mathbf{X}))$ propagates through the PDE dynamics similarly as in (5). Finally, we summarize the Bayesian inverse model for $a(\mathbf{x})$ as follows:

$$
\begin{aligned}
\tilde{\mathbf{Y}}|\tilde{u}(\mathbf{X}), \tilde{a}(\mathbf{X}), \tilde{\mathbf{h}} \sim \text{q-ED}_{\tilde{N}}(\tilde{\mathbf{h}}, \tilde{\boldsymbol{\Gamma}}), \quad \tilde{\boldsymbol{\Gamma}} = \nabla\tilde{P}(\tilde{\mathbf{u}}_0, \tilde{\mathbf{a}}_0)\begin{bmatrix} \mathbf{S}_u & \mathbf{0} \\ \mathbf{0} & \mathbf{S}_a \end{bmatrix}\nabla\tilde{P}(\tilde{\mathbf{u}}_0, \tilde{\mathbf{a}}_0)^{\mathsf{T}} + \delta\mathbf{I}_{\tilde{N}}, \\
\tilde{u} \sim \text{q-}\mathcal{EP}(0, \tilde{\mathcal{C}}_u), \quad \tilde{a} \sim \text{q-}\mathcal{EP}(0, \tilde{\mathcal{C}}_a).
\end{aligned}
\tag{10}
$$

where $\tilde{\mathbf{a}}_0$ can be similarly chosen as $\tilde{\mathbf{a}}_{n-1}$ from the previous training epoch or simply $\tilde{\mathbf{a}}$. The variational Bayes procedure in Section 3.2 can be modified accordingly to obtain the variational solution of $\tilde{a}(\mathbf{X})|\tilde{\mathbf{Y}}$. Meanwhile, we obtain the variational solution of $\tilde{u}(\mathbf{X})|\tilde{\mathbf{Y}}$ as a byproduct.

# 4 Convergence Theorem

In this section, we study the posterior contraction of the Bayesian model (7) in the infinite data limit. Similar theory can be developed for the model (10). We focus on $q \in [1, 2]$ and leave the technically more challenging case $q \in (0, 1)$ to future study. For brevity, we denote $u = \tilde{u}$, $\mathcal{C} = \tilde{\mathcal{C}}$, and $n = N$.

Consider the separable Banach space $\mathbb{X} = (L^q(\Omega), \|\cdot\|_q) \supset (B^{s,q}(\Omega), \|\cdot\|_{s,q})$ for $s > d\left(\frac{2}{q} - \frac{1}{2}\right)$, and $\mathbb{Y} = \mathbb{R}^n$. Define the concentration function of Q-EP measure $\Pi$ at $u = u^\dagger$ as

$$\varphi_{u^\dagger}(\varepsilon) = \inf_{h \in B^{s,q}(\Omega): \|h - u^\dagger\|_q \le \varepsilon} \frac{1}{2} \|h\|_{s,q}^q - \log \Pi(\|u\|_q \le \varepsilon). \tag{11}$$

Let $\mathbb{P}_u^{(n)}$ be the measure of the observations $\mathbf{Y}^{(n)}$ on $(\mathbb{Y}, \mathcal{B}, \mu_0)$ having density $p_u$ and corresponding potential function $\Phi(u; \cdot)$ with respect to the Lebesgue measure $\mu_0$, i.e. $\frac{d\mathbb{P}_u^{(n)}}{d\mu_0}(\mathbf{Y}) = p_u(\mathbf{Y}) \propto \exp(-\Phi(u; \mathbf{Y}))$. Define the Hellinger distance as $d_{n,H}^2(u, u') = \int (\sqrt{p_u} - \sqrt{p_{u'}})^2 d\mu_0$. We have the following posterior contraction theorem.

**Theorem 4.1** (Posterior Contraction). *Let $u \sim$ q-$\mathcal{EP}(0, \mathcal{C})$ with $\mathcal{C}$ satisfying Assumption 1 in $\Theta := L^q(\Omega)$ and $\mathbb{P}_u^{(n)}$ is the measure of $\mathbf{Y}^{(n)}$ parameterized by $u$ with PDE (4) satisfying Assumption 2. If the true value $u^\dagger \in \Theta$ is in the support of $u$, and $\varepsilon_n$ satisfies the rate equation $\varphi_{u^\dagger}(\varepsilon_n) \le n\varepsilon_n^2$ with $\varepsilon_n \ge n^{-\frac{1}{2}}$, then there exists $\Theta_n \subset \Theta$ such that $\Pi_n(u \in \Theta_n : d_{n,H}(u, u^\dagger) \ge M_n\varepsilon_n | \mathbf{Y}^{(n)}) \to 0$ in $P_{u^\dagger}^{(n)}$-probability for every $M_n \to \infty$.*

*Proof.* See Appendix B. $\qquad\square$

Denote $a \wedge b = \min\{a, b\}$, $a \vee b = \max\{a, b\}$, and $x_+ = x \vee 0$. By solving the inequality $\varphi_{u^\dagger}(\varepsilon_n) \le n\varepsilon_n^2$ for the minimal $\varepsilon_n$, we obtain the posterior contraction rate as follows.

**Theorem 4.2** (Contraction Rate). *Let $u \sim$ q-$\mathcal{EP}(0, \mathcal{C})$ with $\mathcal{C}$ satisfying Assumption 1 in $\Theta := L^q(\Omega)$. The rest of the settings are the same as in Theorem 4.1. If the true value $u^\dagger \in B^{s^\dagger, q^\dagger}(\Omega)$ with $s^\dagger > s' + \left(\frac{d}{q^\dagger} - \frac{d}{q}\right)_+$, $s' = \frac{d}{q} - \frac{d}{2}$, and $q^\dagger, q \in [1, 2]$, then we have the rate of the posterior contraction as $\varepsilon_n = n^{-\frac{\sigma(s,q,s^\dagger,q^\dagger) - s'}{2(\sigma(s,q,s^\dagger,q^\dagger) - s') + q(s - \sigma(s,q,s^\dagger,q^\dagger))}}$, where $\sigma(s, q, s^\dagger, q^\dagger) = \left(s - \frac{d}{q}\right) \wedge \left(s^\dagger - \left(\frac{d}{q^\dagger} - \frac{d}{q}\right)_+\right)$.*

*Proof.* See Appendix B. $\qquad\square$

**Remark 3.** *If we set the smoothness parameter $s = s^\dagger + \frac{d}{q} - \left(\frac{d}{q^\dagger} - \frac{d}{q}\right)_+$, and allow the integrability parameter $q \le q^\dagger$, then the contraction rate $\varepsilon_n$ is maximized as $\varepsilon_n^\dagger = n^{-\frac{1}{2 + \frac{d}{s^\dagger - s'}}} > n^{-\frac{1}{2}}$. Note that this optimal rate is achieved regardless of the value of modeling regularization parameter $q$ as long as $q \le q^\dagger$. This implies that when modeling inhomogeneous data and derivative information, under-smoothing (with smaller regularization parameter $q$) is preferred to over-smoothing. When the true integrability $q^\dagger$ is at least $L_1$, setting $q = 1$ guarantees the fastest convergence of the posterior. In the following section, we will demonstrate such optimal choice with various numerical examples.*

# 5 Numerical Experiments

In this section, we test variational Q-EP for solving and learning PDEs. Using the Eikonal equation, Burgers' equation, a nonlinear elliptic equation (Section C.3), and an inverse problem involving the Darcy flow, we investigate the proposed method for a variety of different $q$s. We include PINN [37] and B-PINN [49] as baselines and the variational GP as a special case with $q = 2$ and mainly compare them using the relative error of the estimated solution $\hat{u}$ with reference to the true solution $u^\dagger$: RLE-$p = \frac{\|u^\dagger - \hat{u}\|_p}{\|u^\dagger\|_p}$ for $p = 1, 2, \infty$. Q-EP with $q = 1$ outperforms PINN and variational GP in all cases and attains the best results in most comparisons. The numerical evidence also supports that Q-EP converges the fastest with the optimal choice of $q = 1$.

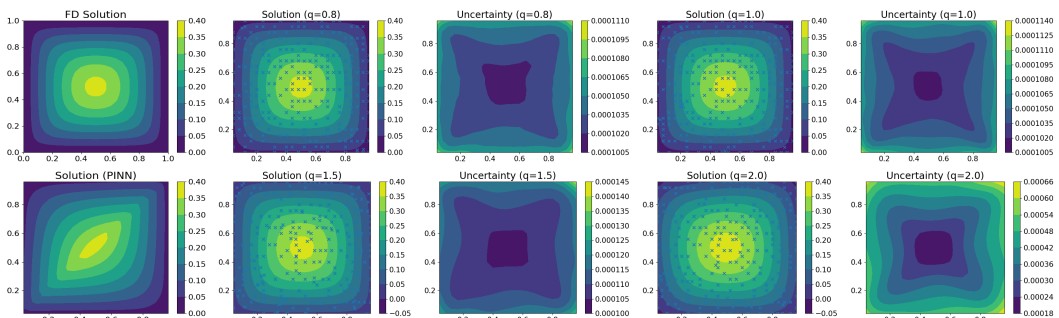

Figure 2: Solving Eikonal equation (12) using high-resolution finite difference method (upper left), PINN (lower left), Q-EPs (middle two in upper row: $q = 0.8$; right two in upper row: $q = 1.0$; middle two in lower row: $q = 1.5$), and GP (right two in lower row: $q = 2.0$) respectively. Blue crosses are learned inducing points.

In most experiments, we choose the interior collocation points on a $24 \times 24$ mesh grid and the corresponding 100 boundary collocation points unless otherwise stated. For the sparse variational inference, $M = 256$ inducing points are randomly initialized and learned by optimizing the ELBO (9). The kernel $\mathcal{C}$ of Q-EP/GP is chosen to be `matern52` with the hyperparameters, e.g. the correlation strength, automatically tuned in the python package `GPyTorch` [11] implemented based on `PyTorch`. PINN is configured to have neural network parameters (weights and biases) of similar size to the variational Q-EP model. The computer codes are publicly available at `https://github.com/lanzithinking/Diff_QEP`.

## 5.1 Eikonal Equation

First, we consider the following regularized Eikonal equation on $\Omega = [0, 1]^2$ also considered in [6]:

$$|\nabla u(\mathbf{x})|^2 - \varepsilon \Delta u(\mathbf{x}) = f(\mathbf{x})^2, \quad \mathbf{x} \in \Omega, \\ u(\mathbf{x}) = 0, \quad \mathbf{x} \in \partial\Omega. \tag{12}$$

where $f \equiv 1$ and $\varepsilon = 0.1$. Based on the set-up in Section 3.1, we define the nonlinear function $D(u, d_1 u, d_2 u, d_1^2 u, d_2^2 u) = (d_1 u)^2 + (d_2 u)^2 - \varepsilon(d_1^2 u + d_2^2 u)$. Then the observations are $\mathbf{Y} = P(\tilde{u}(\mathbf{X})) = \begin{bmatrix} D(u(\mathbf{X}_d), \frac{\partial}{\partial x_1} u(\mathbf{X}_d), \frac{\partial}{\partial x_2} u(\mathbf{X}_d), \frac{\partial^2}{\partial x_1^2} u(\mathbf{X}_d), \frac{\partial^2}{\partial x_2^2} u(\mathbf{X}_d)) \\ u(\mathbf{X}_b) \end{bmatrix}$, and $\mathbf{h} = \begin{bmatrix} f(\mathbf{X}_d)^2 \\ \mathbf{0}_{N_b} \end{bmatrix}$. We obtain $N_d = 24^2$ interior and $N_b = 100$ boundary collocation points. Then we apply the variational Bayes in Section 3.2 to solve (12) with $M = 256$ inducing points. Though not required for convergence, we train each algorithm for 5000 iterations for fair comparison (PINN and B-PINN need much more training epochs than Q-EP solvers to converge).

Figure 2 compares solutions and uncertainty estimates generated by a variety of Q-EP solvers for $q = 0.8, 1.0, 1.5$ and $2.0$ (GP) respectively. We solve the equation using a highly-resolved finite difference method with the Cole-Hopf transformation [6] and use it as the true solution for comparison (upper left). We notice that the solution by PINN (lower left) is much worse and UQ is not available. All Q-EP solvers ($q < 2$) produce better solutions than GP ($q = 2$) which does not precisely characterize the pyramid feature of the true solution. Only two solvers with $q = 0.8$ and $1.0$ yield solutions that correctly match the range of true solution. GP also manifests higher uncertainty in its generated solution. Table C.1 further verifies the superior accuracy of Q-EP solvers compared to GP and PINN (B-PINN) in terms of multiple error metrics including MAE, MSE and RLE by repeating the experiments for 10 times with different random seeds. In this example, Q-EP solver with $q = 1.0$ attains the result comparable to the most accurate solution. Note that with similar size of collocation points, our best results (4.43e-2 in $L_2$ error and 1.03e-2 in $L_\infty$ error) are comparable to those reported (1.64e-2 in $L_2$ error and 7.76e-2 in $L_\infty$ error) in [6] from which UQ is absent.

## 5.2 Burgers' Equation

Next, we test our Q-EP solvers on the Burgers' equation [6] with $\nu = 0.1$:

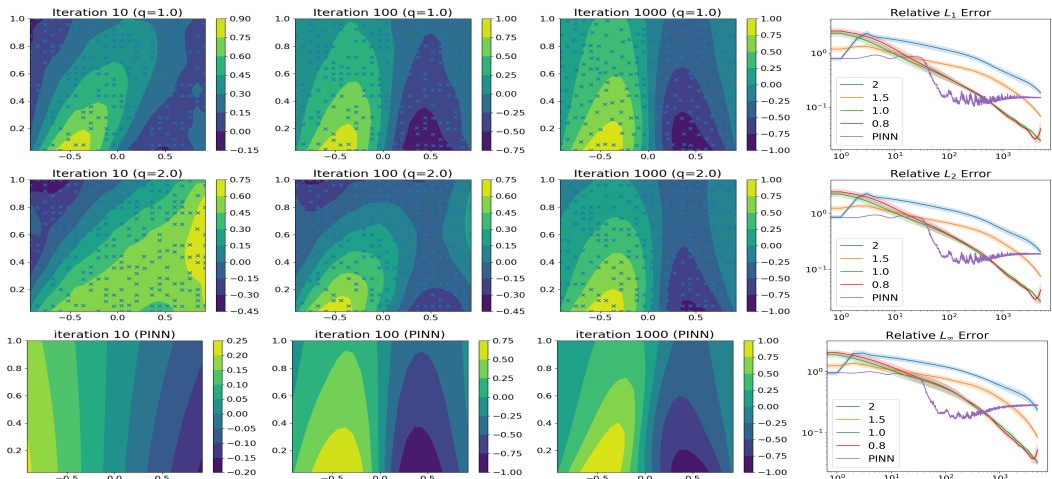

Figure 3: Comparing convergence of Q-EP (left three in top row: $q = 1.0$), GP (left three in middle row: $q = 2.0$) and PINN (left three in bottom row) in solving Burgers' equation (13), with the right column illustrating the error reducing in $L_1$ norm (top), $L_2$ norm (middle), and $L_\infty$ norm (bottom) respectively. Blue crosses are learned inducing points. Shaded regions are standard errors based on 10 repeated experiments.

$$\frac{\partial}{\partial t}u + u\frac{\partial}{\partial x}u - \nu\frac{\partial^2}{\partial x^2}u = 0, \quad (x, t) \in (-1, 1) \times (0, 1],$$
$$u(x, 0) = -\sin(\pi x), \quad x \in (-1, 1), \tag{13}$$
$$u(-1, t) = u(1, t) = 0, \quad t \in (0, 1].$$

We use the same experiment setup as above. Figure C.1 compares the solutions by a highly-resolved finite difference method (upper left, treated as true solution for comparison purpose), PINN (lower left), Q-EP (right three in upper row: $q = 1.0$), and GP (right three in lower row: $q = 2.0$). Q-EP is about one order of magnitude more accurate than PINN and GP, which can be verified from the pointwise error plots and Table 1. The plots of posterior standard deviation on the right column also indicate meaningful uncertainty in the middle area around $x = 0$ where the shock is difficult to resolve. Q-EP still achieves one order of magnitude lower uncertainty compared with GP.

Table 1: Comparing accuracy of various solvers for Burgers' equation (13) in terms of mean absolute error (MAE), mean squared error (MSE), and relative errors in $L_1$ norm (RLE-1), $L_2$ norm (RLE-2), and $L_\infty$ norm (RLE-$\infty$) respectively. Result in each cell are averaged over 10 experiments with different random seeds; values after $\pm$ are standard deviations of these repeated experiments.

| Model ($q$) | MAE | MSE | RLE-1 | RLE-2 | RL-$\infty$ |
|---|---|---|---|---|---|
| PINN | 5.81e-2 $\pm$ 1.03e-4 | 7.12e-3 $\pm$ 3.96e-4 | 0.1508 $\pm$ 0.0017 | 0.1896 $\pm$ 0.0052 | 0.2842 $\pm$ 0.0071 |
| B-PINN | 2.94e-2 $\pm$ 1.57e-2 | 1.67e-3 $\pm$ 1.58e-3 | 0.0785 $\pm$ 0.0420 | 0.0833 $\pm$ 0.0409 | 0.1327 $\pm$ 0.0446 |
| 0.5 | 7.77e-2 $\pm$ 7.09e-2 | 3.50e-2 $\pm$ 7.15e-2 | 0.2018 $\pm$ 0.3751 | 0.2056 $\pm$ 0.3764 | 0.2177 $\pm$ 0.3838 |
| 0.8 | 1.56e-2 $\pm$ 4.96e-3 | 1.59e-3 $\pm$ 5.25e-3 | 0.0405 $\pm$ 0.0780 | 0.0434 $\pm$ 0.0804 | 0.0529 $\pm$ 0.0924 |
| 1.0 | **9.33e-3** $\pm$ 1.76e-4 | **1.77e-4** $\pm$ 2.16e-4 | **0.0242** $\pm$ 0.0128 | **0.0266** $\pm$ 0.0140 | **0.0324** $\pm$ 0.0145 |
| 1.2 | 1.87e-2 $\pm$ 3.23e-4 | 5.92e-4 $\pm$ 4.06e-4 | 0.0485 $\pm$ 0.0151 | 0.0522 $\pm$ 0.0168 | 0.0560 $\pm$ 0.0181 |
| 1.5 | 2.64e-2 $\pm$ 8.58e-4 | 1.31e-3 $\pm$ 1.23e-3 | 0.0684 $\pm$ 0.0262 | 0.0754 $\pm$ 0.0316 | 0.0854 $\pm$ 0.0430 |
| 2.0(Gaussian) | 7.13e-2 $\pm$ 8.68e-3 | 1.08e-2 $\pm$ 1.28e-2 | 0.1848 $\pm$ 0.0908 | 0.2068 $\pm$ 0.1118 | 0.2376 $\pm$ 0.1456 |
| 2.5 | 2.26e-1 $\pm$ 3.20e-1 | 2.49e-1 $\pm$ 6.04e-1 | 0.5879 $\pm$ 0.8296 | 0.6669 $\pm$ 0.9576 | 0.7558 $\pm$ 1.0521 |

Table 1 compares the accuracy of solutions in terms of MAE, MSE and RLE, for which Q-EP with $q = 1.0$ attains the most accurate one, verifying its best performance. The error for Q-EP models does not decrease monotonically with $q > 0$, but reaches the lowest at $q = 1.0$. To further investigate the convergence, we plot solutions by Q-EP (top row: $q = 1.0$), GP (middle row: $q = 2.0$), and PINN (bottom row) in Figure 3 at 10 (first column), 100 (second column), and 1000 (third column) iterations respectively. We can tell that Q-EP with $q = 1.0$ converges the fastest to the best estimate, which is confirmed by the error-reducing plots on the right column. Note that in this example, our algorithm does not need random perturbation of mesh grid as required by [6].

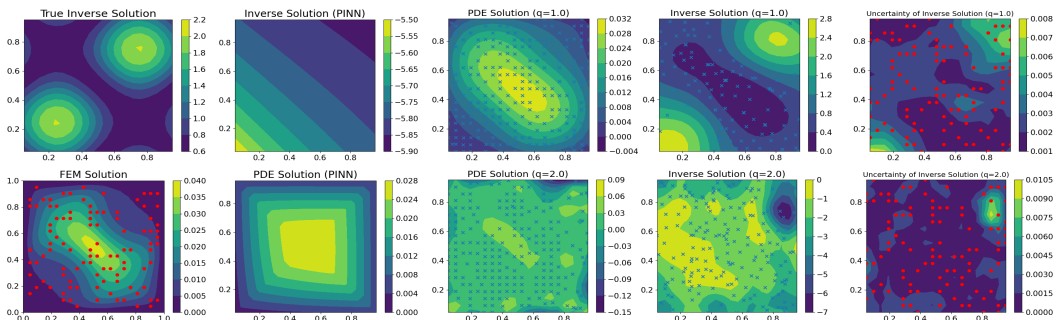

Figure 4: Solving inverse Darcy flow (14) using PINN (second column), Q-EP (right three in upper row: $q = 1.0$), and GP (right three in lower row: $q = 2.0$) respectively. Upper left: true inverse solution $a^\dagger$; lower left: fine-resolution finite element solution $u^\dagger$ to (14) with $a^\dagger$. Blue crosses are learned inducing points, and red dots indicate locations of observations.

## 5.3 Inverse Darcy Flow

Now we consider an inverse problem that involves the following Darcy flow:

$$
\begin{aligned}
-\mathrm{div}(\exp(a)\nabla u)(\mathbf{x}) &= f(\mathbf{x}), \quad \mathbf{x} \in \Omega := [0, 1]^2, \\
u(\mathbf{x}) &= 0, \quad \mathbf{x} \in \partial\Omega.
\end{aligned}
\tag{14}
$$

where the true coefficient such that $\exp(a^\dagger(\mathbf{x})) = \exp(\sin(2\pi x_1) + \sin(2\pi x_2)) + \exp(-\sin(2\pi x_1) - \sin(2\pi x_2))$ [6] is plotted in the upper left panel of Figure 4. We generate data by solving (14) with $a^\dagger$ on a mesh $80 \times 80$ to obtain $u^\dagger$ using the finite element method, illustrated in the lower left panel in Figure 4. On a (coarser) mesh $N_d = 20 \times 20$ used for inference, we randomly select $N_o = 100$ points $\mathbf{X}_o$ in $\Omega$ and obtain observations as $u^\dagger(\mathbf{X}_o) + \varepsilon$ with noise $\varepsilon \sim N(0, \gamma^2 \mathbf{I}_{N_o})$ for $\gamma = 10^{-3}$.

We solve the inverse problem of finding $a$ in (14) given these observations. With $N_d = 400$ interior and $N_b = 84$ boundary collocation points and $M = 256$ inducing points, we train Q-EP solvers for 2000 iterations. Figure 4 illustrates the forward PDE solution $u$ (third column), the inverse solution $a$ (forth column), and the uncertainty of $a$ (rightmost column). Compared with the true $a^\dagger$, Q-EP ($q = 1.0$) recovers $a$ more faithfully than GP ($q = 2.0$) and PINN (upper in the second column). Meanwhile, Q-EP also generates the solution $u$ much closer to that by the finite element method, as shown in the lower left panel. Higher uncertainty is observed by Q-EP ($q = 1.0$) around the corners with less data (Figure C.3), reflecting the configuration of observations. If run for longer (e.g. 5000) iterations, PINN may improve its forward solution, but still gets a poor inverse solution (Figure C.4).

Table C.5 compares the relative errors (RLEs) of forward and inverse solutions. Q-EP with $q = 1.0$ also achieves the best or comparable solutions. Here we emphasize that all the results are based on only $N_o = 100$ observations from one solution, as opposed to the thousands of PDE solutions typically used in operator learning algorithms. Figure C.5 compares the inverse solutions of Q-EP with those of GP in increasingly finer meshes. If we view training in finer mesh as a process to see more data, Q-EP ($q = 1.0$) has already converged fast to better estimates in coarser mesh, leaving less room to improve compared to GP ($q = 2.0$).

## 6 Conclusion

In this paper, we propose variational Q-EP to solve and learn PDEs. We advocate Q-EP with $q = 1.0$ over GP ($q = 2.0$) for modeling derivative information and hence a better probabilistic solver for PDEs. The fastest convergence at $q = 1.0$ is theoretically justified and empirically verified using two nonlinear forward PDE problems and an inverse Darcy flow problem.

One of the limitations might be the variational inference adopted in this paper. The highly nonlinear nature of some PDEs imposes challenges on the quality of variational approximation to the resulting posterior, which in turn may undermine both the solution accuracy and the associated UQ. A possible remedy could be more flexible inference methods, such as normalizing flow [41, 36].

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

# Technical Appendices for "Solving and Learning Partial Differential Equations with Variational Q-Exponential Process"

## A  Calculations

Table A.1: The structure of kernel $\tilde{\mathcal{C}}$ with derivatives.

| $\mathrm{Cov}(\cdot,\cdot)$ | $u(\mathbf{x}')$ | $\frac{\partial}{\partial \mathbf{x}'} u(\mathbf{x}')$ | $\frac{\partial^2}{\partial (\mathbf{x}')^2} u(\mathbf{x}')$ |
|---|---|---|---|
| $u(\mathbf{x})$ | $\mathcal{C}(\mathbf{x},\mathbf{x}')$ | $\frac{\partial}{\partial \mathbf{x}'}\mathcal{C}(\mathbf{x},\mathbf{x}')$ | $\frac{\partial^2}{\partial (\mathbf{x}')^2}\mathcal{C}(\mathbf{x},\mathbf{x}')$ |
| $\frac{\partial}{\partial \mathbf{x}} u(\mathbf{x})$ | $\frac{\partial}{\partial \mathbf{x}}\mathcal{C}(\mathbf{x},\mathbf{x}')$ | $\frac{\partial^2}{\partial \mathbf{x}\partial \mathbf{x}'}\mathcal{C}(\mathbf{x},\mathbf{x}')$ | $\frac{\partial^3}{\partial \mathbf{x}\partial (\mathbf{x}')^2}\mathcal{C}(\mathbf{x},\mathbf{x}')$ |
| $\frac{\partial^2}{\partial \mathbf{x}^2} u(\mathbf{x})$ | $\frac{\partial^2}{\partial \mathbf{x}^2}\mathcal{C}(\mathbf{x},\mathbf{x}')$ | $\frac{\partial^3}{\partial \mathbf{x}^2\partial \mathbf{x}'}\mathcal{C}(\mathbf{x},\mathbf{x}')$ | $\frac{\partial^4}{\partial \mathbf{x}^2\partial (\mathbf{x}')^2}\mathcal{C}(\mathbf{x},\mathbf{x}')$ |

### A.1  Variational Lower Bound

Because $\log p(\mathbf{Y}) = \mathrm{KL}(q(\tilde{\mathbf{U}})\|p(\tilde{\mathbf{U}}|\mathbf{Y})) + \mathcal{L}(q(\tilde{\mathbf{U}}))$, it reduces to maximizing the lower bound $\mathcal{L}(q(\tilde{\mathbf{U}}))$. The sparse variational approximation [46, 47] is adopted by introducing inducing points $\tilde{\mathbf{X}} \in \mathbb{R}^{M\times d}$ with their function values $\tilde{\mathbf{V}} = \tilde{u}(\tilde{\mathbf{X}}) \in \mathbb{R}^{M\times D}$. Hence the joint distribution of $\mathbf{Y}$ and $\tilde{\mathbf{U}}$ can be augmented by including $\tilde{\mathbf{V}}$:

$$p(\mathbf{Y},\tilde{\mathbf{U}}) \propto p(\mathbf{Y}|\tilde{\mathbf{U}})p(\tilde{\mathbf{U}}|\tilde{\mathbf{V}},\mathbf{X},\tilde{\mathbf{X}})p(\tilde{\mathbf{V}}|\tilde{\mathbf{X}}),$$

where we have $\mathrm{vec}(\tilde{\mathbf{V}})|\tilde{\mathbf{X}} \sim \text{q-ED}_{MD}(\mathbf{0},\tilde{\mathbf{C}}_{MM})$ and the conditional distribution

$$\mathrm{vec}(\tilde{\mathbf{U}})|\tilde{\mathbf{V}},\mathbf{X},\tilde{\mathbf{X}} \sim \text{q-ED}_{ND}(\tilde{\mathbf{C}}_{NM}\tilde{\mathbf{C}}_{MM}^{-1}\mathrm{vec}(\tilde{\mathbf{V}}),\tilde{\mathbf{C}}_{NN}-\tilde{\mathbf{C}}_{NM}\tilde{\mathbf{C}}_{MM}^{-1}\tilde{\mathbf{C}}_{MN}).$$

Now we approximate the joint posterior $p(\tilde{\mathbf{U}},\tilde{\mathbf{V}}|\mathbf{Y})$ with the following variational distribution

$$q(\tilde{\mathbf{U}},\tilde{\mathbf{V}}) = p(\tilde{\mathbf{U}}|\tilde{\mathbf{V}})q(\tilde{\mathbf{V}}), \quad q(\tilde{\mathbf{V}}) \sim \text{q-ED}_{MD}(\boldsymbol{\mu},\boldsymbol{\Sigma}),$$

where the covariance $\boldsymbol{\Sigma}$ is of size $MD \times MD$ and can be chosen as a (block)-diagonal matrix for convenience. A standard variational Bayes procedure yields the variational bound:

$$
\log p(\mathbf{Y}) \geq \int q(\tilde{\mathbf{U}},\tilde{\mathbf{V}}) \log \frac{p(\mathbf{Y}|\tilde{\mathbf{U}})p(\tilde{\mathbf{U}}|\tilde{\mathbf{V}},\mathbf{X})p(\tilde{\mathbf{V}})}{q(\tilde{\mathbf{U}},\tilde{\mathbf{V}})} d\tilde{\mathbf{U}}d\tilde{\mathbf{V}}
$$

$$
= \int p(\tilde{\mathbf{U}}|\tilde{\mathbf{V}})q(\tilde{\mathbf{V}})d\tilde{\mathbf{V}} \log p(\mathbf{Y}|\tilde{\mathbf{U}})d\tilde{\mathbf{U}} + \int q(\tilde{\mathbf{V}}) \log \frac{p(\tilde{\mathbf{V}})}{q(\tilde{\mathbf{V}})} d\tilde{\mathbf{V}}
$$

$$
= \mathbb{E}_{q(\tilde{\mathbf{U}})} \log p(\mathbf{Y}|\tilde{\mathbf{U}}) - \mathrm{KL}(q(\tilde{\mathbf{V}})\|p(\tilde{\mathbf{V}})).
$$

where the marginal variational distribution $q(\tilde{\mathbf{U}})$ can be obtained as [17, 30]

$$q(\tilde{\mathbf{U}}) = \int q(\tilde{\mathbf{U}},\tilde{\mathbf{V}})d\tilde{\mathbf{V}} = \int p(\tilde{\mathbf{U}}|\tilde{\mathbf{V}})q(\tilde{\mathbf{V}})d\tilde{\mathbf{V}} \sim \text{q-ED}(\tilde{\mathbf{u}},\mathbf{S})$$

$$\tilde{\mathbf{u}} = \tilde{\mathbf{C}}_{NM}\tilde{\mathbf{C}}_{MM}^{-1}\mathrm{vec}(\boldsymbol{\mu}), \quad \mathbf{S} = \tilde{\mathbf{C}}_{NN} + \tilde{\mathbf{C}}_{NM}\tilde{\mathbf{C}}_{MM}^{-1}(\boldsymbol{\Sigma}-\tilde{\mathbf{C}}_{MM})\tilde{\mathbf{C}}_{MM}^{-1}\tilde{\mathbf{C}}_{MN}.$$

Denote by $\varphi(r;\mathbf{C},N) := -\frac{1}{2}\log|\mathbf{C}| + \frac{N}{2}\left(\frac{q}{2}-1\right)\log r - \frac{1}{2}r^{\frac{q}{2}}$ which is convex for $q \in (0,2]$. Let $\varphi_0(r) = \varphi(r;\boldsymbol{\Gamma},N)$ and $r(\mathbf{Y}) = (\mathbf{Y}-\mathbf{h})^{\mathsf{T}}\boldsymbol{\Gamma}^{-1}(\mathbf{Y}-\mathbf{h})$ be a quadratic form of random variable $\mathbf{Y}$. Then $\log p(\mathbf{Y}|\tilde{\mathbf{U}}) = \varphi_0(r(\mathbf{Y}))$. Therefore, by Jensen's inequality, we can bound from below as

$$\langle \log p(\mathbf{Y}|\tilde{\mathbf{U}})\rangle_{q(\tilde{\mathbf{U}})} = \langle \varphi_0(r(\mathbf{Y}))\rangle_{q(\tilde{\mathbf{U}})} \geq \varphi_0(\langle r(\mathbf{Y})\rangle_{q(\tilde{\mathbf{U}})}), \quad \langle r(\mathbf{Y})\rangle_{q(\tilde{\mathbf{U}})} = \tilde{\mathbf{u}}^{\mathsf{T}}\boldsymbol{\Gamma}^{-1}\tilde{\mathbf{u}} + \mathrm{tr}(\boldsymbol{\Gamma}^{-1}\mathbf{S}).$$

Now we compute the K-L divergence $\mathrm{KL}_{\tilde{\mathbf{V}}} := \mathrm{KL}(q(\tilde{\mathbf{V}})\|p(\tilde{\mathbf{V}}))$:

$$\mathrm{KL}_{\tilde{\mathbf{V}}} = \int q(\tilde{\mathbf{V}}) \log q(\tilde{\mathbf{V}})d\tilde{\mathbf{V}} - \int q(\tilde{\mathbf{V}}) \log p(\tilde{\mathbf{V}})d\tilde{\mathbf{V}} = -\mathcal{H}_q(\tilde{\mathbf{V}}) - \langle \log p(\tilde{\mathbf{V}})\rangle_{q(\tilde{\mathbf{V}})}.$$

Denote by $r(\tilde{\mathbf{V}}) = \text{vec}(\tilde{\mathbf{V}} - \boldsymbol{\mu})^{\mathsf{T}} \boldsymbol{\Sigma}^{-1} \text{vec}(\tilde{\mathbf{V}} - \boldsymbol{\mu})$. Then $\log q(\tilde{\mathbf{V}}) = \varphi(r(\tilde{\mathbf{V}}); \boldsymbol{\Sigma}, MD)$. From [Proposition A.1. of 24] we know that $r^{\frac{q}{2}} \sim \chi^2(MD)$. Therefore

$$\mathcal{H}_q(\tilde{\mathbf{V}}) = \frac{1}{2} \log|\boldsymbol{\Sigma}| + \frac{MD}{2}\left(\frac{q}{2} - 1\right)\frac{2}{q}\mathcal{H}(\chi^2(MD)) + \frac{MD}{2}$$

$$= \frac{1}{2}\log|\boldsymbol{\Sigma}| + \frac{MD}{2}\left(1 - \frac{2}{q}\right)\left[\frac{MD}{2} + \log\left(2\Gamma\left(\frac{MD}{2}\right)\right) + \left(1 - \frac{MD}{2}\right)\psi\left(\frac{MD}{2}\right)\right] + \frac{MD}{2}.$$

Let $\varphi_1(r) := \varphi(r; \tilde{\mathbf{C}}_{MM}, MD)$. Then by Jensen's inequality

$$\langle \log p(\tilde{\mathbf{V}})\rangle_{q(\tilde{\mathbf{V}})} = \langle\varphi_1(\text{vec}(\tilde{\mathbf{V}})^{\mathsf{T}}\tilde{\mathbf{C}}_{MM}^{-1}\text{vec}(\tilde{\mathbf{V}}))\rangle_{q(\tilde{\mathbf{V}})} \geq \varphi_1(\langle\text{vec}(\tilde{\mathbf{V}})^{\mathsf{T}}\tilde{\mathbf{C}}_{MM}^{-1}\text{vec}(\tilde{\mathbf{V}})\rangle_{q(\tilde{\mathbf{V}})}),$$

$$\langle\text{vec}(\tilde{\mathbf{V}})^{\mathsf{T}}\tilde{\mathbf{C}}_{MM}^{-1}\text{vec}(\tilde{\mathbf{V}})\rangle_{q(\tilde{\mathbf{V}})} = \langle\text{vec}(\boldsymbol{\mu})^{\mathsf{T}}\tilde{\mathbf{C}}_{MM}^{-1}\text{vec}(\boldsymbol{\mu})\rangle + \text{tr}(\boldsymbol{\Sigma}\tilde{\mathbf{C}}_{MM}^{-1}).$$

Therefore, the final evidence lower bound (ELBO) $\mathcal{L}^*(q(\tilde{\mathbf{U}}))$ is

$$\log p(\mathbf{E}) \geq \mathcal{L}^*(q(\tilde{\mathbf{U}})) = \varphi(\tilde{\mathbf{u}}^{\mathsf{T}}\boldsymbol{\Gamma}^{-1}\tilde{\mathbf{u}} + \text{tr}(\boldsymbol{\Gamma}^{-1}\mathbf{S}); \boldsymbol{\Gamma}, N)$$

$$+ \frac{1}{2}\log|\boldsymbol{\Sigma}| + \varphi(\langle\text{vec}(\boldsymbol{\mu})^{\mathsf{T}}\tilde{\mathbf{C}}_{MM}^{-1}\text{vec}(\boldsymbol{\mu})\rangle + \text{tr}(\boldsymbol{\Sigma}\tilde{\mathbf{C}}_{MM}^{-1}); \tilde{\mathbf{C}}_{MM}, MD).$$

## B  Proofs

**Notations**: $\lesssim$ means "less than or approximately equal to"; $a_n \lesssim b_n$ implies $a_n \leq C b_n$ for some constant $C > 0$. $\asymp$ means "asymptotically equal to"; $a_n \asymp b_n$ implies $\lim_{n\to\infty} \frac{a_n}{b_n} = c$ for some constant $c$.

*Proof of Proposition 2.1.* Note $r(\tilde{u}_\ell)^{\frac{q}{2}} = \lambda_\ell^{-\frac{q}{2}}|\tilde{u}_\ell|^q \sim \chi^2(1)$ for all $\ell \in \mathbb{N}$ by Proposition A.1. of [24]. Denote $\chi_\ell^2 := \lambda_\ell^{-\frac{q}{2}}|\tilde{u}_\ell|^q \overset{iid}{\sim} \chi^2(1)$. Hence $\|\tilde{u}\|_q^q = \sum_{\ell=1}^\infty \lambda_\ell^{\frac{q}{2}}\chi_\ell^2$ becomes an infinite mixture of chi-squared random variables whose density is analytically intractable. Yet we have

$$\mathbb{E}[\|\tilde{u}(\cdot)\|_q^q] = \sum_{\ell=1}^\infty \lambda_\ell^{\frac{q}{2}}\mathbb{E}[\chi_\ell^2] = \sum_{\ell=1}^\infty \lambda_\ell^{\frac{q}{2}} < \infty,$$

if Assumption 1 holds. Thus it completes the proof.  $\square$

*Proof of Proposition 3.1.* Based on (5) and (6), the potential $\Phi(\tilde{u}; \mathbf{Y}) = -\varphi(r; \boldsymbol{\Gamma}, N)$ where $\varphi(r; \boldsymbol{\Gamma}, N) := -\frac{1}{2}\log|\boldsymbol{\Gamma}| + \frac{N}{2}\left(\frac{q}{2} - 1\right)\log r - \frac{1}{2}r^{\frac{q}{2}}$ is convex in $r$ if $q \in (0, 2]$, and $r = r(\tilde{\mathbf{u}})$ with

$$r(\tilde{\mathbf{u}}) = (\mathbf{A}\tilde{\mathbf{u}} - \mathbf{b})^{\mathsf{T}}\boldsymbol{\Gamma}^{-1}(\mathbf{A}\tilde{\mathbf{u}} - \mathbf{b}), \quad \mathbf{A} = \nabla P(\tilde{\mathbf{u}}_0), \quad \mathbf{b} = -P(\tilde{\mathbf{u}}_0) + \nabla P(\tilde{\mathbf{u}}_0)\tilde{\mathbf{u}}_0 + \mathbf{h}, \quad \boldsymbol{\Gamma} = \mathbf{A}\mathbf{S}\mathbf{A}^{\mathsf{T}} + \delta\mathbf{I}.$$

Since convex function is Lipschitz continuous over compact domain, it suffices to prove that $r(\tilde{\mathbf{u}})$ is bounded (both bounds achievable) and Lipschitz.

Note that $r(\tilde{\mathbf{u}}) \leq \delta^{-1}\|\mathbf{A}\tilde{\mathbf{u}} - \mathbf{b}\|_2^2 \leq \delta^{-1}(\|\mathbf{A}\|\|\tilde{\mathbf{u}}\| + \|\mathbf{b}\|)^2$ where $\tilde{\mathbf{u}}$ represents a PDE solution in the compact domain $\overline{\Omega} \subset \mathbb{R}^d$ and is therefore bounded by some $M > 0$. By Assumption 2, $\|\mathbf{A}\| \leq C$. Therefore, $0 \leq r(\tilde{\mathbf{u}}) \leq \delta^{-1}(CM + \|\mathbf{b}\|)^2$.

On the other hand, we have the gradient of the quadratic form $r(\tilde{\mathbf{u}})$ bounded as

$$\|\nabla r(\tilde{\mathbf{u}})\| = 2\|\mathbf{A}^{\mathsf{T}}\boldsymbol{\Gamma}^{-1}(\mathbf{A}\tilde{\mathbf{u}} - \mathbf{b})\| \leq 2\delta^{-1}\|\mathbf{A}\|\|\mathbf{A}\tilde{\mathbf{u}} - \mathbf{b}\| \leq 2\delta^{-1}C(CM + \|\mathbf{b}\|).$$

Hence $r(\tilde{\mathbf{u}})$ is Lipschitz.

Lastly, there exists $L_1, L_2 > 0$ such that

$$|\Phi(\tilde{u}_1; \mathbf{Y}) - \Phi(\tilde{u}_2; \mathbf{Y})| = |\varphi(r(\tilde{\mathbf{u}}_1); \boldsymbol{\Gamma}, N) - \varphi(r(\tilde{\mathbf{u}}_2); \boldsymbol{\Gamma}, N)|$$
$$\leq L_1|r(\tilde{\mathbf{u}}_1) - r(\tilde{\mathbf{u}}_2)| \leq L_1 L_2\|\tilde{\mathbf{u}}_1 - \tilde{\mathbf{u}}_2\|_q$$
$$\leq 2L_1 L_2\left(\frac{N}{|\Omega|}\right)^{\frac{1}{q}}\|\tilde{u}_1 - \tilde{u}_2\|_q$$

assuming the collocation points are uniformly sampled. Therefore, it completes the proof.  $\square$

According to Theorem 3.1 and Lemma 5.14 of [1], the following general contraction conditions hold for Q-EP and will be used in the proof of posterior contraction Theorem 4.1.

**Theorem B.1.** *Let $\mu$ be a* q-$\mathcal{EP}(0,\mathcal{C})$ *measure satisfying Assumption 1 in the separable Banach space $(L^q(\Omega), \|\cdot\|_q)$, where $q \in [1,2]$. Let $u \sim \mu$ and the true parameter $u^\dagger \in L^q(\Omega)$. Assume $\varepsilon_n > 0$ such that $\varphi_{u^\dagger}(\varepsilon_n) \leq n\varepsilon_n^2$, where $n\varepsilon_n^2 \gtrsim 1$. Then for any $C > 1$, there exists a measurable set $B_n \subset L^q(\Omega)$ and a constant $R > 0$ depending on $C$ and $q$, such that*

$$\log N(4\varepsilon_n, B_n, \|\cdot\|_q) \leq Rn\varepsilon_n^2, \tag{15}$$

$$\mu(u \notin B_n) \leq \exp(-Cn\varepsilon_n^2), \tag{16}$$

$$\mu(\|u - u^\dagger\|_{s',q} < 2\varepsilon_n) \geq \exp(-n\varepsilon_n^2), \tag{17}$$

*where $N(4\varepsilon_n, B_n, \|\cdot\|_q)$ is the minimal number of $\|\cdot\|_q$-balls of radius $4\varepsilon_n$ to cover $B_n$.*

We need the following lemma to bound the Hellinger distance, Kullback-Leibler (K-L) divergence, and K-L variation to complete the proof of Theorem 4.1 [21].

**Lemma B.1.** *Suppose the potential function $\Phi$ (6) satisfies Lipschitz continuity in $u$ as in Proposition 3.1. Then we have*

- $d_H(p_u, p_{u'}) \lesssim \|u - u'\|_q$.

- $K(p_u, p_{u'}) \lesssim \|u - u'\|_q$.

- $V(p_u, p_{u'}) \lesssim \|u - u'\|_q^2$.

*Proof.* First, we consider K-L divergence:

$$K(p_u, p_{u'}) = \int \log \frac{p_u}{p_{u'}} p_u d\mu = \int (\Phi(u'; y) - \Phi(u; y)) p_u d\mu(y) \leq L\|u - u'\|_q$$

by Proposition 3.1. Similarly, we have for K-L variation:

$$V(p_u, p_{u'}) = \int \left(\log \frac{p_u}{p_{u'}}\right)^2 p_u d\mu = \int |\Phi(u'; y) - \Phi(u; y)|^2 p_u d\mu(y) \leq L^2 \|u - u'\|_q^2.$$

Lastly, we bound the Hellinger distance:

$$2d_H^2(p_u, p_{u'}) = \int (\sqrt{p_u} - \sqrt{p_{u'}})^2 d\mu = \int \left[1 - \exp\left(\frac{1}{2}\Phi(u; y) - \frac{1}{2}\Phi(u'; y)\right)\right]^2 p_u d\mu(y)$$

$$\leq \int \frac{C}{4} |\Phi(u'; y) - \Phi(u; y)|^2 p_u d\mu(y) \leq \frac{CL^2}{4} \|u - u'\|_q^2.$$

where the inequality holds for $\|u - u'\|_q^2$ small enough. $\square$

*Proof of Theorem 4.1.* Based on [Theorem 1 of 12], it suffices to verify the following two conditions (the entropy condition (2.4), and the prior mass condition (2.5)) for some universal constants $\eta, K > 0$ and sufficiently large $k \in \mathbb{N}$,

$$\sup_{\varepsilon > \varepsilon_n} \log N(\eta\varepsilon/2, \{u \in \Theta_n : d_{n,H}(u, u^\dagger) < \varepsilon\}, d_{n,H}) \leq n\varepsilon_n^2, \tag{18}$$

$$\frac{\Pi_n(u \in \Theta_n : k\varepsilon_n < d_{n,H}(u, u^\dagger) \leq 2k\varepsilon_n)}{\Pi_n(B_n(u^\dagger, \varepsilon_n))} \leq e^{Kn\varepsilon_n^2 k^2/2}, \tag{19}$$

where the left side of (18) is logarithm of the minimal number of $d_{n,H}$-balls of radius $\xi\varepsilon/2$ needed to cover a ball of radius $\varepsilon$ around the true value $u^\dagger$; $B_n(u^\dagger, \varepsilon_n) = \{u \in \Theta : \frac{1}{n}K(u^\dagger, u) \leq \varepsilon^2, \frac{1}{n}V(u^\dagger, u) \leq \varepsilon^2\}$ with $K(u^\dagger, u) = K(p_{u^\dagger}, p_u)$ and $V(u^\dagger, u) = V(p_{u^\dagger}, p_u)$.

Since $u(\cdot) \in L^q(\Omega)$ satisfy conditions for Theorem B.1, there exists $B_n \subset L^q(\Omega)$ such that (15)-(17) holds. Now we set $\Theta_n = B_n$. For $\forall u, u' \in \Theta_n$ such that $\|u(\cdot) - u'(\cdot)\|_q \leq \varepsilon_n$, we have $d_{n,H}(u, u') \lesssim \|u - u'\|_q \leq \varepsilon_n$ by Lemma B.1. Therefore by (15) we have the following global entropy bound holds

$$\log N(\varepsilon_n, \Theta_n, d_{n,H}) \leq Rn\varepsilon_n^2,$$

which is stronger than the local entropy condition (18).

Now by Lemma B.1 and (17), we have

$$\Pi_n(B_n(u^\dagger, \varepsilon_n)) \geq \Pi_n(\|u^\dagger - u\|_q \leq \varepsilon_n^2, \|u^\dagger - u\|_q^2 \leq \varepsilon_n^2) = \Pi_n(\|u^\dagger - u\|_q^q \leq \varepsilon_n^{2q}) \geq e^{-n\varepsilon_n^2}.$$

Then the prior mass condition (19) is satisfied because the numerator is bounded by 1. The proof is hence completed. □

The following lemma studies the small ball probability in the concentration function (11) [21].

**Lemma B.2** (Small ball probability). *Let $\Pi$ be a $q$-$\mathcal{EP}(0,\mathcal{C})$ prior on $B^{s',q}(\Omega)$ with $s' < s - \frac{d}{q}$. Then as $\varepsilon \to 0$, we have*

$$-\log \Pi(\|u\|_{s',q} \leq \varepsilon) \asymp \varepsilon^{-\frac{1}{\frac{s-s'}{d}-\frac{1}{q}}}.$$

*Proof.* We can compute

$$\Pi(\|u\|_{s',q} \leq \varepsilon) = \mathbb{P}\left[\sum_{\ell=1}^{\infty} (\ell^{\tau_q(s') - \tau_q(s)} |u_\ell|)^q \leq \varepsilon^q\right],$$

where $\mathbb{P}$ is the probability measure on the infinite product space $(L^q(\Omega))^\infty$. From the proof of Proposition 2.1 we know $\|u\|_q^q = \sum_{\ell=1}^{\infty} \lambda_\ell^{\frac{q}{2}} \chi_\ell^2$ is an infinite mixture of $\chi^2(1)$ random variables, so the condition of [Theorem 4.2 of 2] is trivially met and we have

$$\log \mathbb{P}\left[\sum_{\ell=1}^{\infty} (\ell^{\tau_q(s') - \tau_q(s)} |u_\ell|)^q \leq \varepsilon^q\right] \asymp \varepsilon^{-\frac{1}{\tau_q(s) - \tau_q(s') - \frac{1}{q}}}.$$

□

The second lemma gives an upper bound of the first term of the concentration function (11) [21].

**Lemma B.3** (Decentering function). *Assume $u^\dagger \in B^{s^\dagger, q^\dagger}(\Omega)$ for some $s^\dagger > s'$ and $q^\dagger \in [1,2]$. Then as $\varepsilon \to 0$, we have the following bounds*

*(i) If $q^\dagger \geq q$, we require $s^\dagger > s'$:*

$$\inf_{h \in B^{s,q}(\Omega): \|h - u^\dagger\|_{s',q} \leq \varepsilon} \|h\|_{s,q}^q \lesssim \begin{cases} 1, & if\ s < s^\dagger \\ (-\log \varepsilon)^{1 - \frac{q}{q^\dagger}}, & if\ s = s^\dagger\ ; \\ \varepsilon^{-\frac{s-s^\dagger}{s^\dagger - s'}(q \wedge q^\dagger)}, & if\ s > s^\dagger \end{cases}$$

*(ii) If $q^\dagger < q$, we require $s^\dagger > s' - \frac{d}{q} + \frac{d}{q^\dagger}$:*

$$\inf_{h \in B^{s,q}(\Omega): \|h - u^\dagger\|_{s',q} \leq \varepsilon} \|h\|_{s,q}^q \lesssim \begin{cases} 1, & if\ s \leq s^\dagger + \frac{d}{q} - \frac{d}{q^\dagger} \\ \varepsilon^{-\frac{\frac{s-s^\dagger}{d} - \frac{1}{q} + \frac{1}{q^\dagger}}{\frac{s^\dagger - s'}{d} + \frac{1}{q} - \frac{1}{q^\dagger}}q}, & if\ s > s^\dagger + \frac{d}{q} - \frac{d}{q^\dagger} \end{cases}.$$

*Proof.* We identify $u^\dagger \in B^{s^\dagger, q^\dagger}$ with $\{u_\ell^\dagger\}_{\ell=1}^\infty \in \ell^{q^\dagger, \tau_{q^\dagger}(s^\dagger)}$. Then we follow [1] to approximate $u^\dagger$ with $h_{1:L} = \{u_\ell^\dagger\}_{\ell=1}^\infty$ where $u_\ell^\dagger \equiv 0$ for all $\ell > L$. Note $h_{1:L} \in \ell^{q, \tau_q(s)}$ for any finite $L \in \mathbb{N}$. Identifying $h_{1:L}$ with $h \in B^{s,q}(\Omega)$, we could get

$$\|h - u^\dagger\|_{s',q}^q = \sum_{\ell=L+1}^{\infty} \ell^{\tau_q(s')q} |u_\ell^\dagger|^q \leq \begin{cases} \|u^\dagger\|_{s^\dagger, q^\dagger}^{q^\dagger} L^{\frac{s'-s^\dagger}{d}q^\dagger}, & if\ q^\dagger = q \\ \|u^\dagger\|_{s^\dagger, q^\dagger}^q L^{\frac{s'-s^\dagger}{d}q}, & if\ q^\dagger > q\ . \\ \|u^\dagger\|_{s^\dagger, q^\dagger, q}^q L^{\left(\frac{s'-s^\dagger}{d} - \frac{1}{q} + \frac{1}{q^\dagger}\right)q}, & if\ q^\dagger < q \end{cases}$$

Therefore, to have $\|h - u^\dagger\|_{s',q} \leq \varepsilon$ we let

$$L \gtrsim \begin{cases} \varepsilon^{-\frac{d}{s^\dagger - s'}}, & if\ q^\dagger \geq q \\ \varepsilon^{-\frac{1}{\frac{s^\dagger - s'}{d} + \frac{1}{q} - \frac{1}{q^\dagger}}}, & if\ q^\dagger < q \end{cases}. \tag{20}$$

On the other hand, the infimum is less than $\|h\|_{s,q}^q$ with above $h$, which can be bounded as follows. If $q^\dagger = q$,

$$\|h\|_{s,q}^q = \sum_{\ell=1}^{L} \ell^{\tau_{q^\dagger}(s)q^\dagger} |u_\ell^\dagger|^{q^\dagger} \leq \begin{cases} \|u^\dagger\|_{s^\dagger,q^\dagger}^{q^\dagger}, & if\ s \leq s^\dagger \\ \|u^\dagger\|_{s^\dagger,q^\dagger}^{q^\dagger} L^{\frac{s-s^\dagger}{d}q^\dagger}, & if\ s > s^\dagger \end{cases}.$$

If $q^\dagger > q$, by similar argument using Hölder inequality,

$$\|h\|_{s,q}^q = \sum_{\ell=1}^{L} \ell^{\tau_q(s)q} |u_\ell^\dagger|^q \leq \begin{cases} C\|u^\dagger\|_{s^\dagger,q^\dagger}^q, & if\ s < s^\dagger \\ \|u^\dagger\|_{s^\dagger,q^\dagger}^q (\log L)^{1-\frac{q}{q^\dagger}}, & if\ s = s^\dagger \\ \|u^\dagger\|_{s^\dagger,q^\dagger}^q L^{\frac{s-s^\dagger}{d}q}, & if\ s > s^\dagger \end{cases}.$$

If $q^\dagger < q$, by similar argument,

$$\|h\|_{s,q}^q = \sum_{\ell=1}^{L} \ell^{\tau_q(s)q} |u_\ell^\dagger|^q \leq \begin{cases} \|u^\dagger\|_{s^\dagger,q^\dagger,q}^q, & if\ s \leq s^\dagger + \frac{d}{q} - \frac{d}{q^\dagger} \\ \|u^\dagger\|_{s^\dagger,q^\dagger,q}^q L^{\left(\frac{s-s^\dagger}{d} - \frac{1}{q} + \frac{1}{q^\dagger}\right)q}, & if\ s > s^\dagger + \frac{d}{q} - \frac{d}{q^\dagger} \end{cases}.$$

Substituting $L$ in (20) to the above equations yields the conclusion. $\qquad\square$

*Proof of Theorem 4.2.* By Lemmas B.2 and B.3, we have the following bounds for the concentration function (11) as $\varepsilon \to 0$, if $q^\dagger \geq q$,

$$\varphi_{u^\dagger}(\varepsilon) \lesssim \begin{cases} 1 + \varepsilon^{-\frac{1}{\frac{s-s'}{d} - \frac{1}{q}}}, & if\ s < s^\dagger \\ (-\log\varepsilon)^{1-\frac{q}{q^\dagger}} + \varepsilon^{-\frac{1}{\frac{s-s'}{d} - \frac{1}{q}}}, & if\ s = s^\dagger \\ \varepsilon^{-\frac{s-s^\dagger}{s^\dagger - s'}(q \wedge q^\dagger)} + \varepsilon^{-\frac{1}{\frac{s-s'}{d} - \frac{1}{q}}}, & if\ s > s^\dagger \end{cases}.$$

For $s \leq s^\dagger$, the bound is dominated by $\varepsilon^{-\frac{1}{\frac{s-s'}{d} - \frac{1}{q}}}$. For the last case, we need to determine a balancing point of $s$ for the two terms by setting their powers equal. The calculation shows that if $s \leq s^\dagger + \frac{d}{q}$, the bound is still dominated by $\varepsilon^{-\frac{1}{\frac{s-s'}{d} - \frac{1}{q}}}$, but otherwise is dominated by $\varepsilon^{-\frac{s-s^\dagger}{s^\dagger - s'}q}$. Therefore, we have

$$\varphi_{u^\dagger}(\varepsilon) \lesssim \begin{cases} \varepsilon^{-\frac{1}{\frac{s-s'}{d} - \frac{1}{q}}}, & if\ s \leq s^\dagger + \frac{d}{q} \\ \varepsilon^{-\frac{s-s^\dagger}{s^\dagger - s'}q}, & if\ s > s^\dagger + \frac{d}{q} \end{cases}.$$

We need to determine minimal $\varepsilon_n$ such that $\varphi_{u^\dagger}(\varepsilon_n) \leq n\varepsilon_n^2$. Hence for $q^\dagger \geq q$,

$$\varepsilon_n \asymp \begin{cases} n^{-\frac{q(s-s')-d}{2q(s-s')+(q-2)d}}, & if\ s \leq s^\dagger + \frac{d}{q} \\ n^{-\frac{s^\dagger - s'}{2(s^\dagger - s')+q(s-s^\dagger)}}, & if\ s > s^\dagger + \frac{d}{q} \end{cases}.$$

Now if $q^\dagger < q$, by similar argument we have the concentration function (11) as $\varepsilon \to 0$

$$\varphi_{u^\dagger}(\varepsilon) \lesssim \begin{cases} 1 + \varepsilon^{-\frac{1}{\frac{s-s'}{d} - \frac{1}{q}}}, & if\ s \leq s^\dagger + \frac{d}{q} - \frac{d}{q^\dagger} \\ \varepsilon^{-\frac{\frac{s-s^\dagger}{d} - \frac{1}{q} + \frac{1}{q^\dagger}}{\frac{s^\dagger - s'}{d} + \frac{1}{q} - \frac{1}{q^\dagger}}q} + \varepsilon^{-\frac{1}{\frac{s-s'}{d} - \frac{1}{q}}}, & if\ s > s^\dagger + \frac{d}{q} - \frac{d}{q^\dagger} \end{cases}.$$

Thus the contraction rate for $q^\dagger < q$ becomes

$$\varepsilon_n \asymp \begin{cases} n^{-\frac{q(s-s')-d}{2q(s-s')+(q-2)d}}, & if\ s \le s^\dagger + \frac{2d}{q} - \frac{d}{q^\dagger} \\ n^{-\frac{s^\dagger - s' + \frac{d}{q} - \frac{d}{q^\dagger}}{2(s^\dagger - s') + q(s - s^\dagger) - (q-2)(\frac{d}{q} - \frac{d}{q^\dagger})}}, & if\ s > s^\dagger + \frac{2d}{q} - \frac{d}{q^\dagger} \end{cases}.$$

Rewriting the equations into one yields the conclusion. $\qquad\square$

## C   More Numerical Results

### C.1   Eikonal Equation

Table C.1: Comparing accuracy of various solvers for Eikonal equation (12) in terms of mean absolute error (MAE), mean squared error (MSE), and relative errors in $L_1$ norm (RLE-1), $L_2$ norm (RLE-2), and $L_\infty$ norm (RLE-$\infty$) respectively. Result in each cell are averaged over 10 experiments with different random seeds; values after $\pm$ are standard deviations of these repeated experiments.

| Model ($q$) | MAE | MSE | RLE-1 | RLE-2 | RL-$\infty$ |
|---|---|---|---|---|---|
| PINN | 1.77e-2 $\pm$ 9.03e-4 | 4.65e-4 $\pm$ 4.08e-5 | 0.1120 $\pm$ 0.0057 | 0.1167 $\pm$ 0.0051 | 0.1354 $\pm$ 0.0061 |
| B-PINN | 1.81e-2 $\pm$ 2.59e-2 | 1.24e-3 $\pm$ 3.52e-3 | 0.1253 $\pm$ 0.1792 | 0.1162 $\pm$ 0.1700 | 0.2143 $\pm$ 0.2193 |
| 0.5 | 1.96e-3 $\pm$ 5.09e-4 | 6.20e-6 $\pm$ 2.87e-6 | 0.0124 $\pm$ 0.0032 | 0.0132 $\pm$ 0.0029 | 0.0387 $\pm$ 0.0067 |
| 0.8 | **1.44e-3** $\pm$ 3.44e-4 | **3.50e-6** $\pm$ 1.28e-6 | **0.0091** $\pm$ 0.0022 | **0.0100** $\pm$ 0.0018 | 0.0339 $\pm$ 0.0074 |
| 1.0 | 1.68e-3 $\pm$ 5.41e-4 | 4.42e-6 $\pm$ 2.26e-6 | 0.0106 $\pm$ 0.0034 | 0.0110 $\pm$ 0.0030 | 0.0321 $\pm$ 0.0071 |
| 1.2 | 1.69e-3 $\pm$ 3.85e-4 | 4.11e-6 $\pm$ 1.52e-6 | 0.0107 $\pm$ 0.0024 | 0.0108 $\pm$ 0.0020 | **0.0284** $\pm$ 0.0057 |
| 1.5 | 1.68e-3 $\pm$ 3.31e-4 | 4.65e-6 $\pm$ 1.35e-6 | 0.0106 $\pm$ 0.0021 | 0.0116 $\pm$ 0.0018 | 0.0398 $\pm$ 0.0080 |
| 2.0 (Gaussian) | 9.64e-3 $\pm$ 1.33e-3 | 1.30e-4 $\pm$ 3.61e-5 | 0.0610 $\pm$ 0.0084 | 0.0612 $\pm$ 0.0087 | 0.1009 $\pm$ 0.0124 |
| 2.5 | 7.71e-2 $\pm$ 3.18e-2 | 8.36e-3 $\pm$ 5.29e-3 | 0.4881 $\pm$ 0.2011 | 0.4668 $\pm$ 0.1743 | 0.4442 $\pm$ 0.1390 |

To verify the systematic superiority of Q-EP with $q = 1.0$ than GP ($q = 2.0$), we extend our comparison to more kernels including the radius basis function (rbf), $\mathcal{C}(\mathbf{x}, \mathbf{x}') = \sigma^2 \exp(-0.5r^2)$, and rational quadratic (rq), $\mathcal{C}(\mathbf{x}, \mathbf{x}') = \sigma^2(1 + \frac{1}{2\alpha}r^2)^{-\alpha}$, $\alpha > 0$, for $r = \sqrt{\sum_{i=1}^{d}(x_i - x_i')^2/\rho_i^2}$ in Table C.2. Within each type of kernel, Q-EP with $q = 1.0$ consistently outperforms GP ($q = 2.0$). This indicates that the advantage of Q-EP over GP is independent of the choice of kernels (as long as they are compared with the same kernel).

Table C.2: Comparing accuracy of Q-EP ($q = 1$) against GP ($q = 2$) solvers with various kernels for Eikonal equation (12) in terms of mean absolute error (MAE), mean squared error (MSE), and relative errors in $L_1$ norm (RLE-1), $L_2$ norm (RLE-2), and $L_\infty$ norm (RLE-$\infty$) respectively. Result in each cell are averaged over 10 experiments with different random seeds; values after $\pm$ are standard deviations of these repeated experiments.

| Model ($q$) | kernel | MAE | MSE | RLE-1 | RLE-2 | RL-$\infty$ |
|---|---|---|---|---|---|---|
| 1.0 | Matern | **1.68e-3** $\pm$ 5.41e-4 | **4.42e-6** $\pm$ 2.26e-6 | **0.0106** $\pm$ 0.0034 | **0.0110** $\pm$ 0.0030 | **0.0321** $\pm$ 0.0071 |
| 2.0 (Gaussian) | Matern | 9.64e-3 $\pm$ 1.33e-3 | 1.30e-4 $\pm$ 3.61e-5 | 0.0610 $\pm$ 0.0084 | 0.0612 $\pm$ 0.0087 | 0.1009 $\pm$ 0.0124 |
| 1.0 | rbf | **4.39e-3** $\pm$ 1.05e-3 | **3.91e-5** $\pm$ 2.52e-5 | **0.0278** $\pm$ 0.0066 | **0.0327** $\pm$ 0.0107 | **0.0648** $\pm$ 0.0324 |
| 2.0 (Gaussian) | rbf | 1.55e-2 $\pm$ 4.97e-4 | 3.07e-4 $\pm$ 2.87e-5 | 0.0982 $\pm$ 0.0031 | 0.0949 $\pm$ 0.0044 | 0.1236 $\pm$ 0.0048 |
| 1.0 | rq | **1.86e-3** $\pm$ 5.40e-4 | **5.51e-6** $\pm$ 3.44e-6 | **0.0118** $\pm$ 0.0034 | **0.0122** $\pm$ 0.0038 | **0.0168** $\pm$ 0.0064 |
| 2.0 (Gaussian) | rq | 3.07-3 $\pm$ 1.70e-3 | 1.89e-5 $\pm$ 2.83e-5 | 0.0194 $\pm$ 0.0107 | 0.0200 $\pm$ 0.0134 | 0.0325 $\pm$ 0.0242 |

### C.2   Burgers' Equation

Define the nonlinear function $D(u, d_1 u, d_2 u, d_1^2 u) = d_2 u + u d_1 u - \nu d_1^2 u$. The observations $\mathbf{Y}$ can then be expressed as $P(\tilde{u}(\mathbf{X})) = \begin{bmatrix} D(u(\mathbf{X}_d), \frac{\partial}{\partial x} u(\mathbf{X}_d), \frac{\partial}{\partial t} u(\mathbf{X}_d), \frac{\partial^2}{\partial x^2} u(\mathbf{X}_d)) \\ u(\mathbf{X}_b) \end{bmatrix}$, and $\mathbf{h} = \begin{bmatrix} \mathbf{0}_{N_d} \\ \mathbf{g} \end{bmatrix}$, where $\mathbf{g}$ is a vector of size $N_b$ whose elements are $-\sin(\pi x)$ or 0 depending on the order of the corresponding elements in $\mathbf{X}_b$. We use the same experiment setup as above.

In Table C.3, we also observe a consistently better performance of Q-EP ($q = 1.0$) compared to GP ($q = 2.0$) in solving Burgers' equation (13) using various kernels.

One can possibly find a GP with fine-tuned kernel, e.g. rational quadratic (rq), to have better (Q-EP with rbf in Table C.2) or matching (Q-EP with Matern52 in Table C.3) results. However, as a

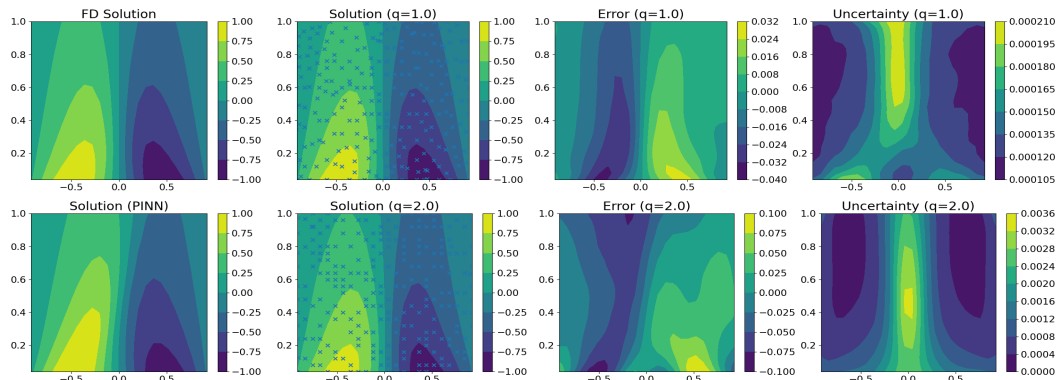

Figure C.1: Solving Burgers' equation (13) using high-resolution finite difference method (upper left), PINN (lower left), Q-EP (right three in upper row: $q = 1.0$), and GP (right three in lower row: $q = 2.0$) respectively. Blue crosses are learned inducing points.

Table C.3: Comparing accuracy of Q-EP ($q = 1$) against GP ($q = 2$) solvers with various kernels for Burgers' equation (13) in terms of mean absolute error (MAE), mean squared error (MSE), and relative errors in $L_1$ norm (RLE-1), $L_2$ norm (RLE-2), and $L_\infty$ norm (RLE-$\infty$) respectively. Result in each cell are averaged over 10 experiments with different random seeds; values after $\pm$ are standard deviations of these repeated experiments.

| Model ($q$) | kernel | MAE | MSE | RLE-1 | RLE-2 | RL-$\infty$ |
|---|---|---|---|---|---|---|
| 1.0 | Matern | **9.33e-3** $\pm$ 1.76e-4 | **1.77e-4** $\pm$ 2.16e-4 | **0.0242** $\pm$ 0.0128 | **0.0266** $\pm$ 0.0140 | **0.0324** $\pm$ 0.0145 |
| 2.0 (Gaussian) | Matern | 7.13e-2 $\pm$ 8.68e-3 | 1.08e-2 $\pm$ 1.28e-2 | 0.1848 $\pm$ 0.0908 | 0.2068 $\pm$ 0.1118 | 0.2376 $\pm$ 0.1456 |
| 1.0 | rbf | **2.51e-3** $\pm$ 1.30e-3 | **1.37e-5** $\pm$ 1.59e-5 | **0.0065** $\pm$ 0.0033 | **0.0074** $\pm$ 0.0040 | **0.0150** $\pm$ 0.0088 |
| 2.0 (Gaussian) | rbf | 2.49e-2 $\pm$ 3.72e-3 | 1.06e-3 $\pm$ 2.63e-4 | 0.0646 $\pm$ 0.0097 | 0.0726 $\pm$ 0.0094 | 0.1144 $\pm$ 0.0073 |
| 1.0 | rq | **2.57e-3** $\pm$ 5.52e-4 | **1.39e-5** $\pm$ 3.15e-6 | **0.0067** $\pm$ 0.0014 | **0.0083** $\pm$ 0.0010 | **0.0199** $\pm$ 0.0049 |
| 2.0 (Gaussian) | rq | 1.09e-2 $\pm$ 1.97e-3 | 1.94e-4 $\pm$ 7.02e-5 | 0.0283 $\pm$ 0.0051 | 0.0309 $\pm$ 0.0057 | 0.0551 $\pm$ 0.0096 |

principled regularization over function spaces, Q-EP facilitates effective modeling of derivatives and differential equations, thereby alleviating the struggle of GP on meticulous kernel engineering.

### C.3 Nonlinear Elliptic Equation

Now we consider a nonlinear elliptic equation with Dirichlet boundary condition on $\Omega = [0, 1]^2$:

$$-\Delta u(\mathbf{x}) + \tau(u(\mathbf{x})) = f(\mathbf{x}), \quad \mathbf{x} \in \Omega,$$
$$u(\mathbf{x}) = g(\mathbf{x}), \quad \mathbf{x} \in \partial\Omega. \tag{21}$$

where we choose $\tau(u) = u^3$, $g(\mathbf{x}) \equiv 0$, and $f$ such that the following true solution $u^\dagger$ satisfies equation (21) [6]

$$u^\dagger(\mathbf{x}) = \sin(\pi x_1)\sin(\pi x_2) + 4\sin(4\pi x_1)\sin(4\pi x_2).$$

We follow the variational procedure in Section 3.1 and use the same experiment setup in Section 5. Figure C.2 compares the solutions by a highly-resolved finite difference method (upper left, treated as true solution), PINN (lower left), Q-EP (right three in upper row: $q = 1.0$), and GP (right three in lower row: $q = 2.0$). Compared to the true solution, Q-EP with $q = 1.0$ yields an estimate more accurate than PINN and GP. Based on Table C.4, the best solution is obtained by Q-EP with $q = 0.5$. Since the true solution is highly fluctuating over the domain $\Omega$, imposing more challenges on the boundary, where the largest pointwise errors occur. Adding more boundary points might help with Q-EP solvers.

### C.4 Inverse Darcy Flow

Finally, we consider a more realistic Darcy flow data used by Fourier Neural Operator (FNO) and Physics-Informed Neural Operator (PINO) available in NVIDIA PhysicsNeMo. This dataset contains thousands of permeability-solution pairs that reflect realistic porous media. Since our method is

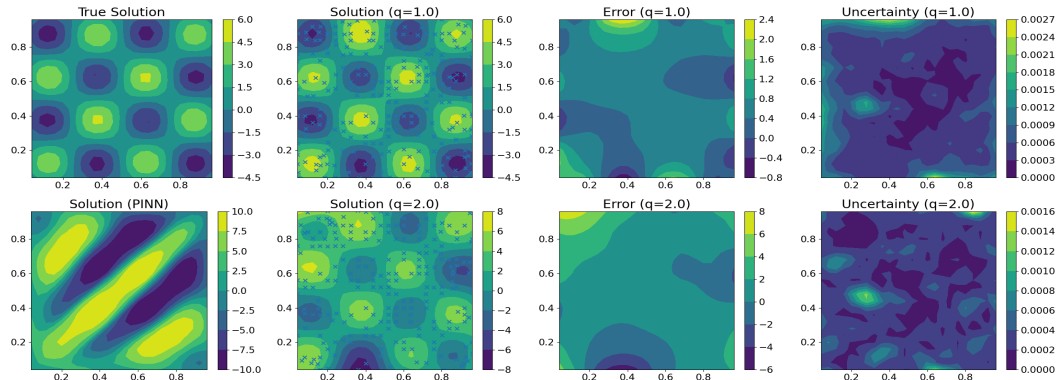

Figure C.2: Solving nonlinear elliptic equation (21) using high-resolution finite difference method (upper left), PINN (lower left), Q-EP (right three in upper row: $q = 1.0$), and GP (right three in lower row: $q = 2.0$) respectively. Blue crosses are learned inducing points.

Table C.4: Comparing accuracy of various solvers for nonlinear elliptic equation (21) in terms of relative error in $L_1$ norm (RLE-1), $L_2$ norm (RLE-2), $L_\infty$ norm (RLE-$\infty$), and time per iteration. Result in each cell are averaged over 10 experiments with different random seeds; values after $\pm$ are standard deviations of these repeated experiments.

| Model ($q$) | RLE-1 | RLE-2 | RLE-$\infty$ | time/iteration |
|---|---|---|---|---|
| PINN | $1.6428 \pm 0.7133$ | $1.5979 \pm 0.6536$ | $1.5595 \pm 0.4462$ | $0.0052 \pm 0.0023$ |
| B-PINN | $0.7026 \pm 0.2879$ | $0.7868 \pm 0.2872$ | $1.2395 \pm 0.3480$ | $0.0108 \pm 0.0013$ |
| 0.5 | $0.0341 \pm 0.0203$ | $0.0514 \pm 0.0322$ | $0.1617 \pm 0.0880$ | $0.2530 \pm 0.0008$ |
| 0.8 | $0.3020 \pm 0.0915$ | $0.3159 \pm 0.0749$ | $0.5549 \pm 0.1092$ | $0.2604 \pm 0.0016$ |
| 1.0 | $0.2835 \pm 0.0532$ | $0.2958 \pm 0.0434$ | $0.5122 \pm 0.0870$ | $0.2614 \pm 0.0005$ |
| 1.2 | $0.3379 \pm 0.0845$ | $0.3342 \pm 0.0689$ | $0.5229 \pm 0.1094$ | $0.2714 \pm 0.0004$ |
| 1.5 | $0.3953 \pm 0.1058$ | $0.4429 \pm 0.1591$ | $0.8938 \pm 0.5714$ | $0.2645 \pm 0.0003$ |
| 2.0(Gaussian) | $0.9876 \pm 0.2777$ | $1.1028 \pm 0.3254$ | $2.0255 \pm 1.1255$ | $0.2562 \pm 0.0004$ |

not to train a surrogate model, we take only one pair and impose the data on $200 \times 200$ mesh to obtain 2000 randomly sampled observations. We then train Q-EP solvers on $60 \times 60$ mesh (with collocation points taken from the grid) and with 512 inducing points. This problem has much larger scale ($\sim 20$ times larger) than all the previous examples. As illustrated in Table C.6, Q-EP with $q = 1$ still achieves remarkable advantage compared with GP ($q = 2$) and PINN. Note, Q-EP with $q = 1$ has comparable accuracy with $q = 0.5$ in forward solution. With only one training pair in the NIVIDA FNO-Darcy data, it is understandably more challenging to obtain the inverse solution, yet for which Q-EP with $q = 1$ attains the best accuracy. In both solutions, GP ($q = 2$) is much worse by most metrics.

Table C.5: Comparing accuracy of various solvers for inverse Darcy flow (14) in terms of relative error in $L_1$ norm (RLE-1), $L_2$ norm (RLE-2), and $L_\infty$ norm (RLE-$\infty$) respectively. Result in each cell are averaged over 10 experiments with different random seeds; values after $\pm$ are standard deviations of these repeated experiments.

| Model ($q$) | Forward PDE Solution | | | Inverse Solution | | |
|---|---|---|---|---|---|---|
| | RLE-1 | RLE-2 | RLE-$\infty$ | RLE-1 | RLE-2 | RLE-$\infty$ |
| PINN | $0.3145 \pm 0.0163$ | $0.3331 \pm 0.0137$ | $0.4225 \pm 0.0250$ | $6.6922 \pm 0.5800$ | $6.2856 \pm 0.5432$ | $4.1070 \pm 0.3189$ |
| 0.5 | $0.2397 \pm 0.0975$ | $0.2689 \pm 0.1137$ | $0.3445 \pm 0.1490$ | $0.4827 \pm 0.2011$ | $0.5217 \pm 0.1916$ | $0.8711 \pm 0.1414$ |
| 0.8 | $0.1669 \pm 0.0242$ | $0.1784 \pm 0.0248$ | $0.2155 \pm 0.0321$ | $\mathbf{0.3577} \pm 0.0758$ | $\mathbf{0.4154} \pm 0.0675$ | $0.8110 \pm 0.1239$ |
| 1.0 | $\mathbf{0.1354} \pm 0.0286$ | $\mathbf{0.1432} \pm 0.0253$ | $\mathbf{0.1810} \pm 0.0193$ | $0.4340 \pm 0.0376$ | $0.4942 \pm 0.0376$ | $0.8494 \pm 0.2132$ |
| 1.2 | $0.1523 \pm 0.0967$ | $0.1549 \pm 0.0898$ | $0.2066 \pm 0.1035$ | $0.5468 \pm 0.1078$ | $0.5921 \pm 0.0890$ | $\mathbf{0.7801} \pm 0.1301$ |
| 1.5 | $0.3270 \pm 0.3257$ | $0.4723 \pm 0.5048$ | $1.2103 \pm 1.0981315$ | $1.6317 \pm 0.8162$ | $1.5914 \pm 0.7810$ | $1.7561 \pm 0.8064$ |
| 2.0(Gaussian) | $0.6328 \pm 0.5954$ | $1.0658 \pm 0.9186$ | $3.1391 \pm 2.2492$ | $2.2096 \pm 0.9100$ | $2.1675 \pm 0.8823$ | $2.8022 \pm 1.6839$ |

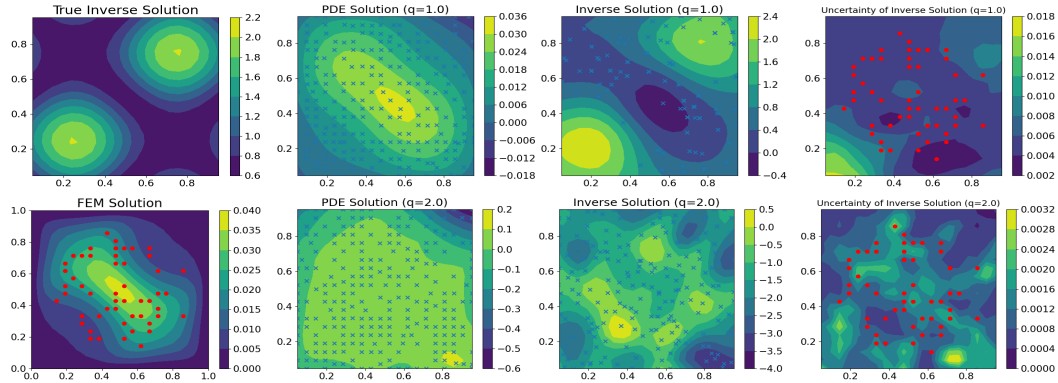

Figure C.3: Solving inverse Darcy flow (14) with sparse data using Q-EP (right three in upper row: $q = 1.0$) and GP (right three in lower row: $q = 2.0$) respectively. Upper left: true inverse solution $a^\dagger$; lower left: fine-resolution finite element solution $u^\dagger$ to (14) with $a^\dagger$. Blue crosses are learned inducing points, and red dots indicate locations of observations.

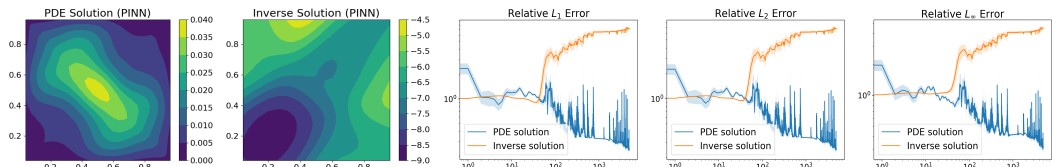

Figure C.4: Solving inverse Darcy flow (14) with PINN running for longer (5000) iterations.

Table C.6: Comparing accuracy of various solvers for the inverse problem with FNO-Darcy data in terms of relative error in $L_1$ norm (RLE-1), $L_2$ norm (RLE-2), and $L_\infty$ norm (RLE-$\infty$) respectively. Result in each cell are averaged over 10 experiments with different random seeds; values after $\pm$ are standard deviations of these repeated experiments.

| Model ($q$) | Forward PDE Solution | | | Inverse Solution | | |
| --- | --- | --- | --- | --- | --- | --- |
| | RLE-1 | RLE-2 | RLE-$\infty$ | RLE-1 | RLE-2 | RLE-$\infty$ |
| PINN | $0.9168 \pm 0.9019$ | $0.8170 \pm 0.7272$ | $0.8190 \pm 0.3859$ | $1.6550 \pm 0.1876$ | $1.5170 \pm 0.1547$ | $1.4303 \pm 0.1190$ |
| 0.5 | $0.3316 \pm 0.0677$ | $0.3191 \pm 0.0402$ | $\mathbf{0.5978} \pm 0.0535$ | $0.7847 \pm 0.1394$ | $0.8548 \pm 0.1683$ | $1.2072 \pm 0.2975$ |
| 1.0 | $0.3239 \pm 0.1297$ | $0.3195 \pm 0.0985$ | $0.6041 \pm 0.0830$ | $\mathbf{0.7724} \pm 0.1798$ | $\mathbf{0.8349} \pm 0.1841$ | $1.2895 \pm 0.2849$ |
| 1.5 | $\mathbf{0.2338} \pm 0.0197$ | $\mathbf{0.2490} \pm 0.0105$ | $0.6149 \pm 0.1056$ | $0.8466 \pm 0.1617$ | $0.8744 \pm 0.1213$ | $1.2070 \pm 0.3958$ |
| 2.0(Gaussian) | $0.5341 \pm 0.5296$ | $0.4904 \pm 0.4292$ | $0.6881 \pm 0.2356$ | $0.9124 \pm 0.1791$ | $0.9251 \pm 0.1502$ | $\mathbf{1.1018} \pm 0.1219$ |

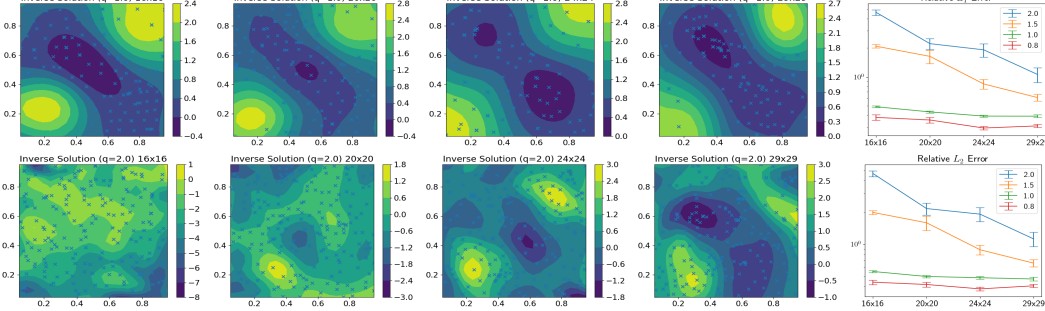

Figure C.5: Comparing solutions to inverse Darcy flow (14) at refined mesh sizes using Q-EP (left four in upper row: $q = 1.0$) and GP (left four in lower row: $q = 2.0$) respectively, with relative errors in $L_1$ norm (upper right) and $L_2$ norm (lower right). Blue crosses are learned inducing points. Error bars indicate standard errors based on 10 repeated experiments with different random seeds.

