# OpenReview forum: "Solving and Learning Partial Differential Equations with Variational Q-Exponential Processes"
_NeurIPS.cc/2025/Conference — NeurIPS 2025 poster_

### Official Review · Reviewer_C3Mw · 2025-06-30

**Clarity:** 2
**Significance:** 3
**Originality:** 3
**Rating:** 4
**Confidence:** 3

**Summary:**

The paper proposes q exponential distributions as an alternative to GPs for solving PDEs with uncertainty quantification. This can be seen as a generalization (q=2 is Gaussian). Specifically for derivative information (which is very relevant for PDEs), the method is claimed as more accurate. The method is well presented within existing literature, and modelling PDEs with uncertainty is a relevant problem for the community. The method is validated on some simple toy problems.

**Questions:**

Q1: The q-exponential distribution seems similar to the generalized Gaussian distribution $\propto \exp(-\frac{1}{2}(|x-\mu|/\alpha)^\beta)$, can you please comment on the differences in the context of your paper?

Q2: What is matern52? (line 136)

Q3: Apart from replacing the Gaussian with the q-exponential, in what ways does your work differ from Hamelijnck, Solin et al.?

Minor questions and remarks:
- line 56: preferable
- line 61: An emerging challenge
- lines 85-88. Very long sentence that is not totally understandable
- Fig. 1: labels are too small. Add labels indicating the rows (q=1, q=2, ..). Same for Fig. 2
- (very minor remark) take a look at your references: **B**esov, **G**aussian, "amp;", ..
- line 508: satisfies

**Ethical Concerns:**

["NO or VERY MINOR ethics concerns only"]

**Final Justification:**

My main concern is the incremental improvement over Hamelijnck et al (2024), Physics-Informed Variational State-Space Gaussian Processes (PHYSS-GP): replacing the Gaussian with Q-EP.

**Limitations:**

In the checklist the authors refer to the scientific contributions at the end of the introduction. I could not find a discussion on societal impacts, however, this should be trivial for this kind of theoretical work.

**Quality:**

3

**Strengths And Weaknesses:**

Strengths:
- Extensive exposition on the methods.
- Learning stochastic PDEs from data is a relevant problem, and not trivial.
- The authors focus on more accurate gradients, and show improvements, both theoretically and empirically.

Weaknesses:
- The method is validated on a limited number of toy problems.
- Writing quality is a bit on the low side. This makes the paper a bit harder to read. Some concrete (friendly) suggestions:
line 99: The negative log density of q-ED (eq. 1) yields ..
line 102: Li et al. [24] prove ..
line 126: Li et al. [24, Theorem 3.5] show that ..
line 131: remove the sentence "We often receive .."

---

> ### Author Rebuttal · Authors · 2025-07-29
>
> We thank the reviewer for the precise summary of the strengths of our paper. Regarding the numerical experiments, these nonlinear PDEs serve as standard benchmarks in similar/related works such as Li et al (FNO, 2021) and Chen et al (2021). We have also added a large scale inverse problem with NVIDIA FNO-Darcy dataset (See the above response to Reviewer rFWA) to strengthen the numerical evidences. The adoption of Q-EP to solve and learn PDEs and the successful demonstration of its superiority over GP consist of the major contributions to the field of physics-informed machine learning. We will also correct the typos and improve the wording of certain sentences according to the reviewer's suggestions.
>
> # Questions:
>
> Q1. You are absolutely right. Univariate Q-EP coincides with the exponential power distribution (a.k.a. generalized Gaussian distribution), as shown in Equation (3) of Li et al (2023). The challenge of developing a valid stochastic process out of it lies in how to generalize the univariate random variable to a multivariate random vector based on which a process can be defined (similar to using MVN to define GP). The Kolmogorov extension theorem determines the requirements which are not trivially met by arbitrary multivariate generalization. For example, the multivariate extension by Gomez et al (1998) fails to satisfy the consistency condition. Q-EP with density in Equation (1) satisfies Kolmogorov's conditions. One can refer to Section 3 of Li et al (2023) for more details.
>
> Q2. `matern52` refers to the Matern kernel with degree $\nu=5/2$. That jargon came from `GPyTorch` and slipped in. We will clarify it in the revision.
>
> Q3. This is a very good question! The work by Hamelijnck et al (2024), Physics-Informed Variational State-Space Gaussian Processes (PHYSS-GP) is also a probabilistic PDE solver based on GP and variational inference. We cited it in the *Connection to the literature* paragraph in the introduction and pointed out it is a closely-related work. This work differs from ours in the following aspects:
>
>   * PHYSS-GP builds on GP; while our work uses more general Q-EP that rigorously regularizes derivatives and outperforms GP in solving PDEs - that's the key contribution of our work.
>   * PHYSS-GP is based on spatiotemporal state-space model which is tailored to solving **time-dependent** PDEs; our framework is for general PDEs and can also be adapted using spatiotemporal Q-EP priors. However, it is unclear how PHYSS-GP solves **time-independent** PDEs and what its benefit is.
>   * PHYSS-GP linearizes a non-linear transformation regarding collocation points in its EKS prior; our Q-EP linearizes the non-linear PDE mapping directly to properly propagate variational distribution. These linearizations are different.
>
> Minor questions and remarks:
>
> Thanks for your careful reading and we will fix them in the revision.
>
> # Limitations:
>
> We discussed the limitation of variational inference for UQ in the concluding Section 6. The societal impacts will be realized through advancements in physics-informed machine learning.

---

### Official Review · Reviewer_MsMP · 2025-07-02

**Clarity:** 3
**Significance:** 2
**Originality:** 2
**Rating:** 4
**Confidence:** 3

**Summary:**

This paper extends recent approaches in solving and learning PDEs based on Gaussian processes to a new class of processes called Q-exponential process. The proposed contribution is the more flexible properties for this new process, especially for PDEs that have stiff solutions, which may be impacted by smoothness assumptions coming from standard GPs.

**Questions:**

1) Theory: In Section 4, there are theoretical results justifying this approach. However, while they look impressive, the message of this section is unclear in the main text. How do these theorems relate to the results on Gaussian processes, if there are any corresponding results? In terms of handling solutions that are less smooth, is there any noticeable improvement over GPs in theory?

2) UQ: The main claim against a PINN based approach is that there is UQ available from GP and q-exponential based approaches. Can authors elaborate why such tightly concentrated uncertainty would be useful? Is there any realistic example where the UQ is meaningful, e.g,. depending on the observation locations (e.g. in a sparser area)? Can authors construct such an example?

3) I'm not entirely convinced that by tuning and changing GP kernels, similar results you have obtained with the q-exponential process cannot be obtained. You have fixed the kernel. Can you elaborate and provide additional results to argue whether the improvements you observe remain with different (suitable) kernels?

4) The comparisons are really thin. Many recent approaches like DeepONets are not compared to - there are similar probabilistic approaches, see e.g. *Vadeboncoeur, Arnaud, et al. "Fully probabilistic deep models for forward and inverse problems in parametric PDEs." Journal of Computational Physics 491 (2023): 112369* and references therein.

I think the experimental section needs much more work before this paper is ready for NeurIPS or a similar venue.

**Ethical Concerns:**

["NO or VERY MINOR ethics concerns only"]

**Final Justification:**

The authors addressed my concerns -- by including numerical evidence, which tilted the paper from Borderline reject to borderline accept.

It is not still a "full accept" for me. I find this general direction not very innovative - since the inference techniques are very costly, and scalability of these methods (GPs) in general are problematic, thus I doubt this will become a usual way to solve these PDEs.

**Limitations:**

Yes, very briefly. It could be expanded.

**Quality:**

3

**Strengths And Weaknesses:**

Strengths: This paper is based on a novel idea of using q-exponential processes in the setting of modelling PDEs. This can lead to interesting applications, especially in the settings author mentioned, e.g., PDEs with erratic solutions.

Weaknesses: I list the weaknesses below - please see questions part for further inquiries:

(1) The paper is somewhat an incremental improvement over the previous approaches. The methodology is rather standard, replacing GP with q-exponential process, but deriving a standard variational inference procedure. Therefore, despite this is an interesting extension, I find the whole framework's novelty a bit limited.

(2) Experimental results are insufficient and not very conclusive in my opinion - to justify the whole framework introduced in this paper.

(3) There are some theoretical results - but zero intuition about how to interpret them. Please see my questions.

---

> ### Author Rebuttal · Authors · 2025-07-29
>
> Thanks to the reviewer for acknowledging the novelty of our work as an interesting extension to GP in solving PDEs. Such extension by Q-EP has theoretical and numerical advantages in modeling inhomogeneous objects and derivative information. The novelty lies in the first application of Q-EP to solving and learning PDEs and successful demonstration of its advantages over GP.
>
> Weaknesses:
>
> (1) We admit that we did not invent new inference methods (and it is not our focus). The adoption of sparse variational inference is not standard or straightforward: the unique format of Q-EP density as in Equation (1) prevents many tractable results in ELBO of GP; therefore it requires significant work to overcome these obstacles in deriving computable ELBO for Q-EP. Additionally, as a Bayesian generalization to Chen et~al (2021), which only provides MAP solution, we creatively propose *distribution propagation* (as in Equation (5)) to properly propagate uncertainty (posterior standard deviation) for appropriate UQ, which is not seen in the existing probabilistic PDE solvers. What's more, the posterior contraction theorems applied to solving and learning PDEs represent a tremendous theoretical development. Therefore, we believe our work makes a significant contribution to the physics-informed machine learning and should not be seen merely as a plug-in extension of existing methods.
>
> (2) We have added new experimental results per your request (see below).
>
> (3) We explained the implication of theorems, particularly theorem 4.2, in remark 3. We will elaborate the points in our revision (and below).
>
> # Questions:
>
> 1. Long story short. The take-home message of Theorem 4.2 is that the posterior contraction rate of our Bayesian Q-EP solvers attains maximal value at $q=1$ so Q-EP with $q=1$ converges the fastest and should be preferred to GP which is a special case of Q-EP with $q=2$. More specifically, as long as the true PDE solution is at least $L_1$ regular, Q_EP solver for $q=1$ should be adopted for the fastest convergence. This is illustrated by our numerical examples and particularly in the rightmost column of Figure 3 where Q-EP ($q=1$) reduces the error the fastest in the training process.
>
> 2. In forward (solving) problems, uncertainty reflects the quality of variational inference—specifically, low uncertainty indicates a well-configured set of inducing points. In inverse (learning) problems, uncertainty reflects the informativeness of the observations—low uncertainty suggests the presence of rich and informative data. One can refer to Figure C.3 and Figure C.4 for a comparison on the uncertainty of inverse solution with different amount of observations. The high uncertainty region usually signals interesting phenomena or data insufficiency, such as the middle region of shocks at $x=0$ in Burgers' equation illustrated in the rightmost columns of Figure C.1. This could be informative and useful for downstream applications such as design of experiment (DoE).
>
> 3. A similar question was asked by Reviewer 8z5k as above. Yes, you can possibly find a GP with fine-tuned kernel to beat Q-EP with a poor kernel. The meaning of using Q-EP is to rigorously regularize on function spaces -- this is especially important when it comes to solving PDEs where large derivatives can easily blow up and hurdle the training. Q-EP, compared to GP, imposes stronger regularization with $q<2$ on the derivatives when solving PDE by modeling the function and its derivatives as Q-EPs and hence enjoys a smoother training and faster convergence. To ease your doubts, we have added another comparison with a radius basis function (rbf) kernel, which is kept the same for different models in the following table summarizing the results of solving Burgers' equation. Note, compared to Table 1 with Matern($\nu=5/2$) kernel, the overall results with rbf are improved (maybe due to the solution smoothness being better matched). **Nevertheless, the Q-EP ($q=1$) with Matern($\nu=5/2$) kernel yields higher accuracy than GP ($q=2$) with rbf kernel!** Note, the hyper-parameters of both kernels were automatically optimized by `GPyTorch` in all experiments.
>
> | Model (Burgers-**rbf**) | MAE | MSE | RLE-1 | RLE-2 | RL-$\infty$ |
> | ------------------ | ----- | ------ | ------- | ------- | ------------- |
> | **1.0** | **2.51e-3** $\pm$ 1.30e-3 | **1.37e-5** $\pm$ 1.59e-5 | **0.0065** $\pm$ 0.0033 | **0.0074** $\pm$ 0.0040 | **0.0150** $\pm$ 0.0088 |
> | 1.5 | 3.13e-2 $\pm$ 3.55e-3 | 1.62e-3 $\pm$ 3.42e-4 | 0.0813 $\pm$ 0.0092 | 0.0900 $\pm$ 0.0094 | 0.1268 $\pm$ 0.0115 |
> | 2.0 | 2.49e-2 $\pm$ 3.72e-3 | 1.06e-3 $\pm$ 2.63e-4 | 0.0646 $\pm$ 0.0097 | 0.0726 $\pm$ 0.0094 | 0.1144 $\pm$ 0.0073 |
> | 2.5 | 3.27e-2 $\pm$ 7.98e-3 | 1.74e-3 $\pm$ 9.48e-4 | 0.0850 $\pm$ 0.0207 | 0.0917 $\pm$ 0.0243 | 0.1220 $\pm$ 0.0089 |
>
> 4. Thanks for the references. First, we want to contend that the numerical results are not thin. We have compared various Q-EP with PINN, B-PINN, GP, and Chen et al (2021) using 4 examples with different focuses including accuracy (tables), convergence (Figure 3), UQ (Figures 2, C.1 and more), influence of data amount (Figures C.3 vs C.4) and resolution (Figure C.6) (refer to the appendix). These examples are standard benchmarks in similar works in the area, including (but not limited to) Chen et~al (2021), Meng and Yang (2023), and more recently Hamelijnck et al (2024). Second, we have also added another more realistic inverse problem from NVIDIA FNO-Darcy dataset (See above response to Reviewer rFWA). Last but not the least, our method is **NOT comparable** to neural operator methods including DeepONets, FNO, PINO and many more:
>
>     * Neural operators require thousands of pairs of inputs (functions of interest such as coefficients, boundaries, etc) and outputs (PDE solutions) to train the neural networks. This usually involves a numerical solver to obtain solutions to the PDE with different inputs. On the contrary, our method does **not require** to prepare solutions as training samples. Instead, it models PDE solution as Q-EP and solves the PDE using variational inference --  the posterior mean is the estimate to the PDE solution and the posterior standard deviation quantifies the uncertainty of the solver.
>     * Neural operators are trained to provide **surrogate** models for quick forward analysis or inverse UQ. While Q-EP probabilistic solver can also get inverse solution (with forward solution as a byproduct, see Section 3.3), it does NOT rely on any surrogate model.
>
> Therefore, it would be an apple-to-orange comparison if comparing Q-EP solver with DeepONet. In checking with our motivating work, Chen et al (2021), and the follow-ups, Meng and Yang (2023) and Hamelijnck et al (2024), none of them compared with neural operator methods. Therefore, we will not include any neural operator method such as DeepONet in our comparison though we appreciate the reviewer's suggestion.

---

> > ### Comment · Reviewer_MsMP · 2025-08-02
> >
> > Many thanks for your response. The responses resolve my comments.
> >
> > I still think the kernel aspect could be better explored with a larger study to ensure for every time $q=1$ seems to outperform a GP, one cannot find a GP kernel that matches the solution equally well or better; so I think this aspect for me is inconclusive. RBF kernel would naturally struggle with something like Burgers' equation -- I think more comprehensive study is required to show this framework is really necessary (and cannot be replaced by better kernel design + GPs).

---

> > > ### Author Response · Authors · 2025-08-03
> > > **added comparison on kernels**
> > >
> > > Thank you very much for your positive feedback! We are glad that our responses have resolved your concerns.
> > >
> > > Regarding the kernels, we agree that it helps with the paper by extending the study on more types. Therefore, we have added rational quadratic kernel and thus compare the following three types:
> > >
> > > Let  $r = d(x, x') = \sqrt{ \sum_{d=1}^D (x_d-x'_d)^2/\rho^2_d }$ denote the distance function.
> > >
> > > * Matern($\nu=5/2$):
> > >
> > > $k(r)= \sigma^2(1+\sqrt{5}r+\frac{5}{3}r^2) \exp(-\sqrt{5}r)$
> > >
> > > * Radius basis function (rbf):
> > >
> > > $k(r) = \sigma^2\exp(-0.5r^2)$
> > >
> > > * Rational quadratic (rq):
> > >
> > > $k(r) = \sigma^2 (1+\frac{1}{2\alpha}r^2)^{-\alpha}, \alpha>0$
> > >
> > > In the following tables, we compare the error metrics of solving Eikonal and Burgers' equations respectively using QEP ($q=1.0$) and GP ($q=2.0$) with **different kernels**. Note, all the hyperparemeters of kernels (e.g. $\sigma^2$, $\rho_d$, and $\alpha$) are automatically optimized by `GPyTorch`.
> > >
> > > |Model (Eikonal) | kernel | MAE | MSE | RLE-1 | RLE-2 | RLE-$\infty$ |
> > > | ----------------- | ------- | ------| ------ | ------- | -------| ----------------|
> > > | $q=1.0$ | Matern | **1.68e-3** $\pm$ 5.41e-4 | **4.42e-6** $\pm$ 2.26e-6 | **0.0106** $\pm$ 0.0034 | **0.0110** $\pm$ 0.0030 | **0.0321** $\pm$ 0.0071 |
> > > | $q=2.0$ (Gaussian) | Matern | 9.64e-3 $\pm$ 1.33e-3 | 1.30e-4 $\pm$ 3.61e-5 | 0.0610 $\pm$ 0.0084 | 0.0612 $\pm$ 0.0087 | 0.1009 $\pm$ 0.0124 |
> > > | $q=1.0$ | rbf | **4.39e-3** $\pm$ 1.05e-3 | **3.91e-5** $\pm$ 2.52e-5 | **0.0278** $\pm$ 0.0066 | **0.0327** $\pm$ 0.0107 | **0.0648** $\pm$ 0.0324 |
> > > | $q=2.0$ (Gaussian) | rbf | 1.55e-2 $\pm$ 4.97e-4 | 3.07e-4 $\pm$ 2.87e-5 | 0.0982 $\pm$ 0.0031 | 0.0949 $\pm$ 0.0044 | 0.1236 $\pm$ 0.0048 |
> > > | $q=1.0$ | rq | **1.86e-3** $\pm$ 5.40e-4 | **5.51e-6** $\pm$ 3.44e-6 | **0.0118** $\pm$ 0.0034 | **0.0122** $\pm$ 0.0038 | **0.0168** $\pm$ 0.0064 |
> > > | $q=2.0$ (Gaussian) | rq | 3.07-3 $\pm$ 1.70e-3 | 1.89e-5 $\pm$ 2.83e-5 | 0.0194 $\pm$ 0.0107 | 0.0200 $\pm$ 0.0134 | 0.0325 $\pm$ 0.0242 |
> > >
> > >
> > > |Model (Burgers) | kernel | MAE | MSE | RLE-1 | RLE-2 | RLE-$\\infty$ |
> > > | ----------------- | ------- | ------| ------ | ------- | -------| ----------------|
> > > | $q=1.0$ | Matern | **9.33e-3** $\pm$ 1.76e-4 | **1.77e-4** $\pm$ 2.16e-4 | **0.0242** $\pm$ 0.0128 | **0.0266** $\pm$ 0.0140 | **0.0324** $\pm$ 0.0145 |
> > > | $q=2.0$ (Gaussian) | Matern | 7.13e-2 $\pm$ 8.68e-3 | 1.08e-2 $\pm$ 1.28e-2 | 0.1848 $\pm$ 0.0908 | 0.2068 $\pm$ 0.1118 | 0.2376 $\pm$ 0.1456 |
> > > | $q=1.0$ | rbf | **2.51e-3** $\pm$ 1.30e-3 | **1.37e-5** $\pm$ 1.59e-5 | **0.0065** $\pm$ 0.0033 | **0.0074** $\pm$  0.0040 | **0.0150**  $\pm$ 0.0088 |
> > > | $q=2.0$ (Gaussian) | rbf | 2.49e-2 $\pm$ 3.72e-3 | 1.06e-3 $\pm$ 2.63e-4 | 0.0646 $\pm$ 0.0097 | 0.0726 $\pm$ 0.0094 | 0.1144 $\pm$ 0.0073 |
> > > | $q=1.0$ | rq | **2.57e-3** $\pm$ 5.52e-4 | **1.39e-5** $\pm$ 3.15e-6 | **0.0067** $\pm$ 0.0014 | **0.0083** $\pm$ 0.0010 | **0.0199** $\pm$ 0.0049 |
> > > | $q=2.0$ (Gaussian) | rq | 1.09e-2 $\pm$ 1.97e-3 | 1.94e-4 $\pm$ 7.02e-5 | 0.0283 $\pm$ 0.0051 | 0.0309 $\pm$ 0.0057 | 0.0551 $\pm$ 0.0096 |
> > >
> > > Note, these added results convey the following messages that we believe further support the intrinsic advantages of QEP over GP:
> > >
> > > * Within each type of kernel, QEP with $q=1$ **consistently outperforms** GP ($q=2$). This indicates that the advantages of QEP over GP is independent of the choice of kernels (as long as they are compared with the same kernel).
> > > * In the example of Eikonal equation, **GP with rational quadratic (rq) kernel has better performance than QEP ($q=1$) with rbf kernel**. In the Burgers' equation, **GP with rational quadratic (rq) kernel has matching performance with QEP ($q=1$) with Matern kernel ($\nu=5/2$)**. As we commented before, one can possibly find a GP with fine-tuned kernel to beat Q-EP with a poor kernel, e.g. polynomial or period that does not well fit the problem.
> > > * Q-EP provides a principled regularization over function spaces, facilitating effective modeling of derivatives and differential equations, thereby alleviating the struggle of GP on meticulous kernel engineering.
> > >
> > > We hope our added results help to resolve your last doubt and further convince you that QEP ($q=1$) is a preferable choice over GP ($q=2$) for solving and learning PDEs. If you have other questions, we will be happy to answer.

---

> > > > ### Comment · Reviewer_MsMP · 2025-08-04
> > > >
> > > > thanks for your reply! I am satisfied that this process has use in this sense, and will increase my score.

---

> > > > > ### Author Response · Authors · 2025-08-04
> > > > >
> > > > > Great! Thank you so much for your generous support!

---

### Official Review · Reviewer_rFWA · 2025-07-02

**Clarity:** 4
**Significance:** 3
**Originality:** 3
**Rating:** 4
**Confidence:** 4

**Summary:**

The paper addresses the limitations of neural network and Gaussian Process (GP)-based methods in solving PDEs, particularly for systems with sharp transitions. It introduces the Q-exponential process (Q-EP), a generalisation of GPs that offers more expressive and flexible function estimators in settings involving non-smooth dynamics and derivative information. Using sparse variational inference, the proposed method achieves scalable and accurate PDE solutions with principled uncertainty quantification. Toy experiments on the Eikonal equation, Burgers’ equation, and inverse Darcy flow show that Q-EP outperforms GPs and neural networks in both accuracy and uncertainty estimation.

**Questions:**

Please refer to the Weaknesses section for Questions.

**Ethical Concerns:**

["NO or VERY MINOR ethics concerns only"]

**Final Justification:**

All of my concerns have been addressed by the authors. I am keeping my initial decision for this paper (borderline accept) but raising my confidence score from 2 to 4 for this decision.

**Limitations:**

yes

**Quality:**

3

**Strengths And Weaknesses:**

I quite like the idea of using the Q-exponential process, which can be seen as a generalised version of GPs for solving PDEs, offering a flexible choice of the exponent $q$. It also makes sense to employ sparse variational inference with inducing points to reduce algorithmic complexity when solving the functions. Overall, the proposed method is well-justified and theoretically sound. However, I have some comments and potential limitations below that I hope the authors can address.
- However, a natural limitation arises in selecting the optimal value of $q$ for different settings. I’m curious about the performance when $q>2$, and is $q=1$ always optimal, as suggested by the experiments? Please clarify this point.
- Another limitation I notice is the absence of real-world datasets in the experiments. I suggest the authors incorporate experiments on real datasets -such as physics-informed datasets, time series, or, ideally, image datasets. I would highly appreciate this addition, and it would positively influence my evaluation further.
- Could the authors mention any potential applications of the Q-exponential process beyond solving PDEs and inverse problems? For example, might it be applicable in modelling processes within diffusion models? I’m curious whether such an extension is feasible.

Overall, I lean toward accepting this work, but only weakly, due to two main limitations: the unclear guidance on selecting the exponent $𝑞$, and the lack of experiments on real-world datasets. I will adjust my score based on the authors’ responses to these concerns.

**[After Rebuttal]**: My concerns have been addressed, and I am happy to raise my confidence score in advocating for the acceptance of this paper (from 2 to 4).

---

> ### Author Rebuttal · Authors · 2025-07-25
>
> We are grateful to the reviewer for the appreciation of our work! In fact, we advocate Q-EP with **$\boxed{q=1}$** over GP (corresponding to $q=2$) and other qs for solving and learning PDEs. The reason is that Q-EP converges the fastest at $q=1$, which is the main take-home message of Theorem 4.2. As explained in Remark 3, the posterior convergence attains the maximal rate at $q=1$. This is also verified in our numerical examples and particularly illustrated in the rightmost column of Figure 3, where Q-EP with $q=1$ has the steepest reduction in error in the process of training.
>
> * As mentioned above, we have the optimal choice of $q=1$ from the posterior contraction perspective. Such choice is justified by Theorem 4.2 and the numerical illustrations. We have also added $q=2.5$ per your request, and it is not surprising that its results are worse as shown in the following tables.
>
> | Model (Eikonal) | MAE | MSE | RLE-1 | RLE-2 | RL-$\infty$ |
> | -----------------  | ----- | ------ | ------- | ------- | ------------- |
> | **1.0** | **1.68e-3** $\pm$ 5.41e-4 | **4.42e-6** $\pm$ 2.26e-6 | **0.0106** $\pm$ 0.0034 | **0.0110** $\pm$ 0.0030 | **0.0321** $\pm$ 0.0071 |
> | 2.0 | 9.64e-3 $\pm$ 1.33e-3 | 1.30e-4 $\pm$ 3.61e-5 | 0.0610 $\pm$ 0.0084 | 0.0612 $\pm$ 0.0087 | 0.1009 $\pm$ 0.0124 |
> | 2.5 | 7.71e-2 $\pm$ 3.18e-2 | 8.36e-3 $\pm$ 5.29e-3 | 0.4881 $\pm$ 0.2011 | 0.4668 $\pm$ 0.1743 | 0.4442 $\pm$ 0.1390 |
>
> | Model (Burgers) | MAE | MSE | RLE-1 | RLE-2 | RL-$\infty$ |
> | ------------------ | ----- | ------ | ------- | ------- | ------------- |
> | **1.0** | **9.33e-3** $\pm$ 1.76e-4 | **1.77e-4** $\pm$ 2.16e-4 | **0.0242** $\pm$ 0.0128 | **0.0266** $\pm$ 0.0140 | **0.0324** $\pm$ 0.0145 |
> | 2.0 | 7.13e-2 $\pm$ 8.68e-3 | 1.08e-2 $\pm$ 1.28e-2 | 0.1848 $\pm$ 0.0908 | 0.2068 $\pm$ 0.1118 | 0.2376 $\pm$ 0.1456 |
> | 2.5 | 2.26e-1 $\pm$ 3.20e-1 | 2.49e-1 $\pm$ 6.04e-1 | 0.5879 $\pm$ 0.8296 | 0.6669 $\pm$ 0.9576 | 0.7558 $\pm$ 1.0521 |
>
> * We have now added a more realistic Darcy flow data (publicly available on NVIDIA website, link suppressed due to NIPS policy) used by Fourier Neural Operator (FNO, Google "nvidia FNO darcy") and Physics-Informed Neural Operator (PINO) available in **NVIDIA PhysicsNeMo**.
> This dataset contains thousands of permeability-solution pairs that reflect realistic porous media. Since our method is not to train a surrogate model (see the response to Reviewer MsMP below for elaboration on the difference), we take **only ONE** pair and impose the data on 200x200 mesh to obtain 2000 randomly sampled observations. We then train Q-EP solvers on 60x60 mesh (with collocation points taken from the grid) and with 512 inducing points. This problem has much larger scale (~20 times larger) than all the previous examples. As illustrated in the following table, Q-EP with $q=1$ still achieves remarkable advantage compared with GP ($q=2$) and PINN. Note, Q-EP with $q=1$ has comparable accuracy with $q=0.5$ in forward solution. With only one training pair in the NIVIDA FNO-Darcy data, it is understandably more challenging to obtain the inverse solution, yet for which Q-EP with $q=1$ attains the best accuracy. In both solutions, GP ($q=2.0$) is much worse by most metrics.
>
> | Model ($q$) | RLE-1 (fwd) | RLE-2 (fwd) | RLE-$\infty$ (fwd) | RLE-1 (inv) | RLE-2 (inv) | RLE-$\infty$ (inv) |
> | -------------- | ------------- | -------------- | ---------------------| ------------- | ------------ | -------------------- |
> | PINN | 0.9168 $\pm$ 0.9019 | 0.8170 $\pm$ 0.7272 | 0.8190 $\pm$ 0.3859 | 1.6550 $\pm$ 0.1876 | 1.5170 $\pm$ 0.1547 | 1.4303 $\pm$ 0.1190 |
> | 0.5 | 0.3316 $\pm$ 0.0677 | 0.3191 $\pm$ 0.0402 | **0.5978** $\pm$ 0.0535 | 0.7847 $\pm$ 0.1394 | 0.8548 $\pm$ 0.1683 | 1.2072 $\pm$ 0.2975 |
> | 1.0 | 0.3239 $\pm$ 0.1297 | 0.3195 $\pm$ 0.0985 | 0.6041 $\pm$ 0.0830 | **0.7724** $\pm$ 0.1798 | **0.8349** $\pm$ 0.1841 | 1.2895 $\pm$ 0.2849 |
> | 1.5 | **0.2338** $\pm$ 0.0197 | **0.2490** $\pm$ 0.0105 | 0.6149 $\pm$ 0.1056 | 0.8466 $\pm$ 0.1617 | 0.8744 $\pm$ 0.1213 | 1.2070 $\pm$ 0.3958 |
> | 2.0 | 0.5341 $\pm$ 0.5296 | 0.4904 $\pm$ 0.4292 | 0.6881 $\pm$ 0.2356 | 0.9124 $\pm$ 0.1791 | 0.9251 $\pm$ 0.1502 | **1.1018** $\pm$ 0.1219 |
>
> * Since proposed by Li et-al (2023), Q-EP has been applied to:
>   - Regularization of latent representation (ICLR 2025)
>   - Deep probabilistic models (AABI 2025)
>
> We are glad that the reviewer's suggestion resonates with our ongoing efforts -- yes, we have been working on the diffusion model with Q-exponential noise and achieved favorable preliminary results.
>
> We hope our added results could convince the reviewer that Q-EP is preferable to GP in solving PDEs and promising as a new tool in physics-informed machine-learning.

---

### Official Review · Reviewer_8z5k · 2025-07-23

**Clarity:** 3
**Significance:** 3
**Originality:** 3
**Rating:** 4
**Confidence:** 2

**Summary:**

The authors propose to use q-exponential process to solve and learn PDEs as they are designed to better handle data with abrupt changes and to more accurately model derivative information.

**Questions:**

1. how do we select the optimal q in real problems? it seems q=1 is nearly always the best and the convergence is faster in theory. do you suggest using q=1 for all tasks

2. it seems that the uncertainty provided by q=1 appears more structurally informative than that of the GP solver q=2. can you elaborate on the reason for this?

3. i feel that it may be a bit misleading to say q-ep is designed to handle abrupt changes in data. does it only refer to q<1? can you comment on that? also, is it possible to adjust the kernel in GP to address the same issue rather than use q-ep?

4. q-ep does not have such trctable conditional and predictive distributions as GP. how do you overcome that?

5. how is q-ep compared with Besov process

**Ethical Concerns:**

["NO or VERY MINOR ethics concerns only"]

**Quality:**

3

**Strengths And Weaknesses:**

Strengths:

1. The idea of using Q-EP is quite novel
2. They have great theoretical justification
3. Uncertainty quantification is provided
4. They demonstrate their approach in a range of tasks and show superior performance

Weaknesses:

for highly nonlinear PDEs, the variational approximation to the true posterior distribution might be poor, which could undermine the final solution

---

> ### Author Rebuttal · Authors · 2025-07-25
>
> We thank the reviewer for acknowledging the novelty and strengths of our work. Our **main contribution** is showing that Q-EP (with q=1) is superior to GP (corresponding to q=2) for solving and learning PDEs -- we have demonstrated this point both theoretically and empirically. GP is so popular in (physics-informed) machine learning that it is almost the universal choice for probabilistic PDE solvers. Our argument is that **an easy but better alternative can be offered by Q-EP**.
>
> Weaknesses:
>
> We adopted sparse variational Bayes for the computational consideration. It works decently well for all the non-linear PDEs included in our paper and it is **an** inference method we used. We admit that extremely non-linear systems, like weather forecasting equations, could pose challenges for our proposed Q-EP based PDE solver. Yet such extreme non-linearity challenges all approximate methods, including GP solvers and neural network based solutions. If the true posterior distribution is the main focus, advanced Monte Carlo methods, such as those involving optimal transportation and normalizing flow, could be used to mitigate the issue. This, however, is not the weakness or limitation of Q-EP itself.
>
> # Questions:
> 1. Yes, we advocate Q-EP with $q=1$ as a superior alternative to GP ($q=2$) for solving and learning PDEs. The convergence rate by Theorem 4.2 attains its peak value at $q=1$, which justifies the optimal choice of $q=1$ -- it converges the fastest. This is also verified in our numerical examples and particularly illustrated in the rightmost column of Figure 3, which shows the error reducing as a function of epochs.
> 2. We are unsure about whether the uncertainty by Q-EP with $q=1$ is more structurally informative than that of GP as we did not claim so. Similarly as Q-EP, the posterior by GP solver also indicates the uncertainty mainly comes from the boundary in the Eikonal equation, and GP solver also has higher uncertainty in the shock area in Burgers' equation. We will be happy to discuss on specific examples.
> 3. We refer to $q<2$ when talking about improved capability of handling inhomogeneous objects compared to GP (which corresponds to $q=2$). Actually, Q-EP (with different qs) has a regularization effect on modeling objects through the parameter $q$ - smaller $q$ imposes stronger regularization and hence reconstructs images sharper (See Figure 1 of Li et al (2023)).
> It is true that kernel choice impacts the model performance. For example, Matern kernels generally perform better than radius basis function (rbf) in imaging analysis. So it is possible, but not necessary, that GP with Matern kernel yields better results than Q-EP with rbf (Refer to the response to Reviewer MsMP below for an added comparison using rbf kernel). Ideal kernel is chosen such that the resulting process matches the smoothness of modeled data, which is usually unknown. Q-EP, on the other hand, offers a rigorous regularization on function spaces through the parameter $q$ rather than carefully choosing kernels. Note, all the comparisons in the paper were based on the same kernel `Matern` ($\nu=5/2$) to ensure the fairness in comparing Q-EP with GP.
> 4. Q-EP DOES have tractable conditional and predictive distributions as GP. This is exactly Theorem 3.5 of Li et al (2023) and quoted as Theorem 2.2 in our paper. This is one of the theoretical and numerical advantages of Q-EP, when compared with Besov.
> 5. This is a very good question! It has been answered in Theorem 3.4 of Li et al (2023). Quoted as Theorem 2.1, Q-EP shares the same series representation as Besov process that is originally defined in terms of series. As also mentioned in the introduction, Q-EP can be viewed as a probabilistic definition of Besov and it has tractable posterior prediction and explicit control of correlation strength (through covariance kernel). Numerically, Q-EP outperforms Besov in low dimensions and behaves similarly in higher dimensions in image reconstruction problems (Li et al, 2023). However, directly applying Besov to solving PDE is impeded by theoretical difficulties, e.g. justifying that the differential of Besov is still Besov, how to obtain conditional and prediction, etc.

---

### Author Response · Authors · 2025-08-06
**summary**

We want to sincerely thank all the anonymous reviewers for their careful reading and insightful suggestions. In this paper, we propose Q-exponential process (Q-EP) as a superior alternative to Gaussian process (GP) for solving and learning PDEs. As a principled regularization over function spaces, Q-EP facilitates effective modeling of derivatives and differential equations, and has advantages over GP verified both theoretically and empirically. All the reviewers acknowledged the novelty of Q-EP as an extension to GP for probabilistic PDE solvers and its potential contribution to the physics-informed machine learning.

We have addressed their major concerns that can be summarized as follows:

* The implication of posterior contraction theorem 4.2/ optimal choice of $q$:

    *We emphasize that the take-home message is that the posterior contraction rate attains maximal value at $q=1$ so Q-EP with $q=1$ converges the fastest and should be preferred to GP (which corresponds to a special case with $q=2$). (Refer to responses to Reviewers 8z5k, rFWA, and MsMP). This will be further clarified in the revision.*

* Numerical experiments:

    *We have added a more realistic large-scale inverse problem using NVIDIA-FNO Darcy flow data and several experiments regarding higher Q-EP power and different covariance kernels. (Refer to responses to Reviewers rFWA, MsMP and C3Mw). These new added results will be incorporated and strengthen the numerical evidences.*

* Equivalence/Replaceability of GP with fine-tuned kernels

    *We argue and have numerically demonstrated that the advantage of Q-EP ($q=1$) over GP ($q=2$) in solving PDEs is independent of kernel choice. While admitting that GP with fine-tuned kernels is possible to outperform Q-EP with poor kernels, we think Q-EP contributes as a new tool for probabilistic PDE solvers with systematic and rigorous advantages compared with GP which may struggle with heavy kernel tuning. (Refer to responses to Reviewers 8z5k and MsMP).*

We have also provided point-to-point responses that address many questions regarding Q-EP itself. We hope to convince all the reviewers and area chair of the merit of Q-EP in solving and learning PDEs. Thanks again for all the constructive comments and engaging discussions!

---

### Note · Authors · 2025-08-11

We thank all the reviewers for their constructive comments and engaging discussions. In the final remarks, we clarify and emphasize the main contributions of our paper based on the rebuttal and discussions:

* Q-exponential process (Q-EP) provides a novel and effective probabilistic PDE solver as a superior alternative to Gaussian process (GP).
* The posterior of Q-EP model for solving and learning PDEs converges the fastest at $q=1$ (preferable to GP corresponding to $q=2$), which has been theoretically verified and numerically demonstrated.
* As a principled regularization over function spaces, Q-EP facilitates effective modeling of derivatives and differential equations, thereby alleviating the struggle of GP on meticulous kernel engineering.

We have successfully addressed the concerns raised by reviewers (see below for the rebuttal summary) and will incorporate all the new results and necessary changes into the revision. We look forward to the feedback and final decision by the ACs.

---

### Decision · Program_Chairs · 2025-09-17

**Decision:**

Accept (poster)

**Comment:**

The reviewed work revisits the use of Gaussian processes for the solution of PDEs by replacing the Gaussian with the q-Gaussian. The authors observe that choosing $q=1$ outperforms the Gaussian (q=2) case on multiple different PDEs. The reviewers appreciate the idea of using the q-Gaussian process, although they are concerned about the additional algorithmic complication due to the use of variational approximations. They are also concerned that the work is somewhat incremental compared to prior approaches. Overall this is a borderline case, but I tend to think that the improvements due to $q=1$ are surprising enough to make this work relevant to the NeurIPS community.